# Proteostatic reactivation of the developmental transcription factor TBX3 drives BRAF/MAPK-mediated tumorigenesis

Zhenlei Zhang[1,9], Yufan Wu[1,9], Jinrong Fu[2,9], Xiujie Yu[3], Yang Su[4], Shikai Jia[1], Huili Cheng[1], Yan Shen[3], Xianghui He[5], Kai Ren[6], Xiangqian Zheng[7], Haixia Guan [2] ✉, Feng Rao [4] ✉ & Li Zhao [8] ✉

MAPK pathway-driven tumorigenesis, often induced by BRAF[V600E], relies on epithelial dedifferentiation. However, how lineage differentiation events are reprogrammed remains unexplored. Here, we demonstrate that proteostatic reactivation of developmental factor, TBX3, accounts for BRAF/MAPK-mediated dedifferentiation and tumorigenesis. During embryonic development, BRAF/MAPK upregulates USP15 to stabilize TBX3, which orchestrates organogenesis by restraining differentiation. The USP15-TBX3 axis is reactivated during tumorigenesis, and *Usp15* knockout prohibits BRAF[V600E]-driven tumor development in a Tbx3-dependent manner. Deleting Tbx3 or Usp15 leads to tumor redifferentiation, which parallels their overdifferentiation tendency during development, exemplified by disrupted thyroid folliculogenesis and elevated differentiation factors such as Tpo, Nis, Tg. The clinical relevance is highlighted in that both USP15 and TBX3 highly correlates with BRAF[V600E] signature and poor tumor prognosis. Thus, USP15 stabilized TBX3 represents a critical proteostatic mechanism downstream of BRAF/MAPK-directed developmental homeostasis and pathological transformation, supporting that tumorigenesis largely relies on epithelial dedifferentiation achieved via embryonic regulatory program reinitiation.

Key mediators of embryogenesis are often reactivated during cell transformation and tumorigenesis, leading to epithelial dedifferentiation and acquirement of stem-like characteristics[1–3]. For example, three pluripotency transcriptional factors (TFs) OCT4, NANOG and SOX2 are aberrantly expressed in different types of tumors, and trigger similar molecular reconfiguration of the epigenomic landscape as in Embryonic stem cells (ESCs), suggesting common mechanistic links between embryogenesis and tumorigenesis[4–6]. Besides stemness factors, pro-growth TFs important for developmental growth, such as Yap, can also be reactivated in tumors and become actively pursued therapeutic targets[7]. To better target disease-specific reprogramming of such developmental TFs, it is important to understand mechanisms regulating their abundance in development and in tumorigenesis.

TBX3, a member of the T-box transcription factor family, is a critical development and tumorigenesis regulator[8]. Proper TBX3 expression directs the embryonic formation of hypothalamus-pituitary, the heart, mammary glands, limbs, and lungs[9,10]. Accordingly, loss-of-function mutations in human TBX3 lead to ulnar-mammary syndrome which is characterized by mammary and apocrine glands hypoplasia, as well as defects of the upper limb, areola, dental structures, heart and genitalia[11]. Recently, the oncogenic function of TBX3 draws attention, where TBX3 is often up-regulated in tumorigenesis and promotes tumor cell proliferation and metastasis, especially during BRAF/MAPK induced tumorigenesis[12–16]. Considering the maldevelopment in the absence of TBX3 and the tumorigenic risk of high TBX3 levels, its expression level needs to be tightly controlled. Thus,

critical morphogenic and oncogenic signaling pathways, such as BMPs[17], Shh[18], TGF-β[19], FGFs[20], Wnt/β-catenin[21], as well as BRAF/MAPK pathway, participate in TBX3 expression regulation, mostly at the transcriptional level[12,15]. The ubiquitin-proteasome system (UPS) provides an essential way to control core signaling factor abundance by selective degradation, such that it mediates many processes in embryonic development, also aberrant regulations of UPS-dependent degradation are involved in different pathological processes especially carcinogenesis[22,23]. In the absence of any known TBX3-specific E3 ligase or deubiquitinase (DUB), whether and how it is regulated by the UPS during development or tumorigenesis remains unclear.

Gene mutations of *BRAF*, especially *BRAF^V600E^*, and constitutively activated MAPK cascades drive multiple types of cancers, depriving differentiated features through melanoma genesis and intestine/colon carcinogenesis[24-27]. Dedifferentiation and development of radioactive iodine (RAI) refractory diseases, resulted from decreased thyroid differentiation factors, especially sodium iodide symporter (NIS), is also the leading cause for BRAF^V600E^-related poor prognosis and mortality in thyroid tumorigenesis[28-30]. Even upregulation of pioneer developmental organizers, such as SOX2, KLF4, β-catenin, SNAIL-1, c-MYC, TBX3, BCL6, et al., are reported downstream of BRAF/MAPK-driven tumor, the pathways causing lineage reprogramming are not clear[12,15,30-36]. Developmentally, MAPK pathways transduce critical mitogenic signals downstream of RTK-RAS during embryogenesis, which are essential for lineage progenitor reservoir maintenance and organ growth[37]. Abundant Erk (diphosphorylated Erk1/2, dp-Erk) activation domains correlate with extraembryonic ectoderm, different brain zones, limb buds, and the foregut[38], thus RAS-ERK pathway disturbance correlate with wide spectrum of phenotypes. However, the mechanisms how active BRAF/MAPK intermingles with lineage differentiation programs to influence developmental and oncogenic processes are unexplored[39].

In this study, we identify USP15, as the DUB for TBX3, deubiquitylating TBX3 and antagonizing UPS-dependent degradation. Activated BRAF/MAPK pathway functions partially through USP15-mediated TBX3 stabilization, thus genetic knockout of Usp15 leads to disrupted organ development and tumorigenesis comparable to Tbx3 mutants. Tbx3 and Usp15 deficiency both lead to differentiation augmentation, exemplified with excessive Tpo, Nis, and Tg in thyroid folliculogenesis and regain the same factors in BRAF^V600E^-driven thyroid tumor development. Our findings reveal the proteostasis regulatory network governing TBX3 expression through embryogenesis and tumorigenesis, thus not only shed light on TBX3-regulated cellular events, but also support the resounding mechanistic link between BRAF/MAPK-directed progenitor and cancerous cell fates.

## Results

### USP15 interacts with and stabilizes TBX3

Aberrant expression of TBX3 has been extensively reported in various human cancers, primarily due to transcriptional dysregulation[9]. To determine the significance of UPS-mediated proteostasis regulation in TBX3 abundance control, we first characterize TBX3 turnover at protein level. Cycloheximide (CHX) pulse-chase experiments demonstrate that TBX3 protein level was reduced gradually, while its mRNA levels remained constant (Supplementary Fig. 1a). Cell lines with different molecular backgrounds were included here, such as mouse embryonic fibroblasts (MEFs), BRAF^V600E^-mutated papillary thyroid cancer (PTC) K1 cells and anaplastic thyroid cancer (ATC) 8505 C cells, as well as an estrogen receptor positive (ER + ) breast cancer cell line MCF-7, and KRAS-mutated lung adenocarcinoma cell line A549, and showed different turnover rates. Further treatment with proteasome inhibitor MG132 resulted in significant accumulation of TBX3. TBX3 turnover rates, indicating that TBX3 undergoes UPS-mediated protein degradation (Supplementary Fig. 1a).

To identify regulators of TBX3 stability, affinity purification and mass spectrometry was performed. The experiments revealed that USP15, an important DUB, is a strong TBX3 interaction partner along with the previously characterized HDAC2[14] (Fig. 1a, and Supplementary Data 1). Interaction between TBX3 and USP15 was further confirmed by Co-IP experiments, where HA-tagged USP15 was efficiently pulled-down by Flag-tagged TBX3 in HEK293T cell extracts, and vice versa (Fig. 1b). The binding is highly specific, since USP11, the close family member of USP15, fails to bind with TBX3 (Supplementary Fig. 1b). Endogenous TBX3 and USP15 showed strong binding across different TBX3 expressing cell lines (Fig. 1c), in which interactions mainly occurred in the cytoplasm confirmed by Proximity Ligation Assays (PLA) as well as immunofluorescence (IF) staining (Fig. 1d, Supplementary Fig. 1c and Video 1–4). Moreover, USP15 over-expression increased TBX3 levels in a dose-dependent manner, while USP15 knock-down decreased TBX3 expression, its mRNA levels remained relatively stable (Fig. 1e, f and Supplementary Fig. 1d). This stabilization of TBX3 depends on USP15's DUB activity, since wild-type USP15, but not the USP15^C269A^ mutant, restored TBX3 loss caused by USP15 depletion (Fig. 1g, h), and ectopic USP15^C269A^ lost the function to prolong TBX3 half-life (Fig. 1i, Supplementary Fig. 1e, f). MG132 treatment similarly restored TBX3 loss caused by USP15 knock-down (Fig. 1j). These studies together suggest that the DUB USP15 stabilizes TBX3 likely by counteracting UPS-mediated TBX3 degradation.

### USP15 is a TBX3 deubiquitinase

We next verified whether USP15 targets TBX3 for deubiquitylation. As expected, ectopic expression of wild-type USP15 but not USP15^C269A^ or USP11 reduced the poly-ubiquitylation of TBX3, while USP15 removal augmented TBX3 poly-ubiquitylation (Fig. 2a, b). Importantly, recombinant USP15 could remove the ubiquitin chain from TBX3 in vitro (Fig. 2c). K48-linked and K63-linked poly-ubiquitylation are common ubiquitylation events, wild-type USP15, but not USP15^C269A^, efficiently removed K48-linked poly-ubiquitylation from TBX3, which is the canonical signal that marks proteins for degradation (Fig. 2d). Thus, USP15 governs TBX3 stability via direct deubiquitylation.

Next, we searched for regions in USP15 and TBX3 accounting for their interaction. USP15 is composed of a DUSP domain at the amino terminus, two UBL domains in the middle, and a carboxyl terminus region (Fig. 2e). A series of deletion and immunoprecipitation experiments suggest that TBX3 primarily interacts with the UBL1 of USP15, such that the UBL1 deletion mutant fails to co-immunoprecipitate TBX3 (Fig. 2f). TBX3 normally functions as a transcriptional repressor through the repression domains located on either the amino (R2) or carboxyl terminus (R1). The DNA-binding domain, so called T-box, and the activation domain (A) is located in the center (Fig. 2e). Domain mapping experiments suggest that the activation domain is most important for the physical interaction with USP15, since its deletion [ΔA and Δ(A + R1) TBX3 variants] inhibited TBX3-USP15 interaction (Fig. 2g). In vitro GST pull-down experiment further showed that the UBL1 of USP15 mediates the direct binding between TBX3, thus ΔU1 USP15 mutant lost the capability to remove poly-ubiquitin chain from TBX3 (Fig. 2h, i). Therefore, specific functional domains within USP15 and TBX3 mediate the direct molecular binding, which is also important for USP15 to deubiquitylate TBX3 (Fig. 2j).

### USP15-TBX3 axis mediates BRAF^V600E^-induced tumorigenesis

TBX3 participates in versatile regulations during tumorigenesis[9], so is USP15[40-42]. We thus wonder whether USP15 functions in tumor formation partially through its regulation of TBX3 expression. Consistent with this notion, USP15 over-expression promoted proliferation of tumor cells (Supplementary Fig. 2a-c). Whereas, knock-down of USP15 dramatically inhibited tumor cell proliferation evidenced by CCK8 and

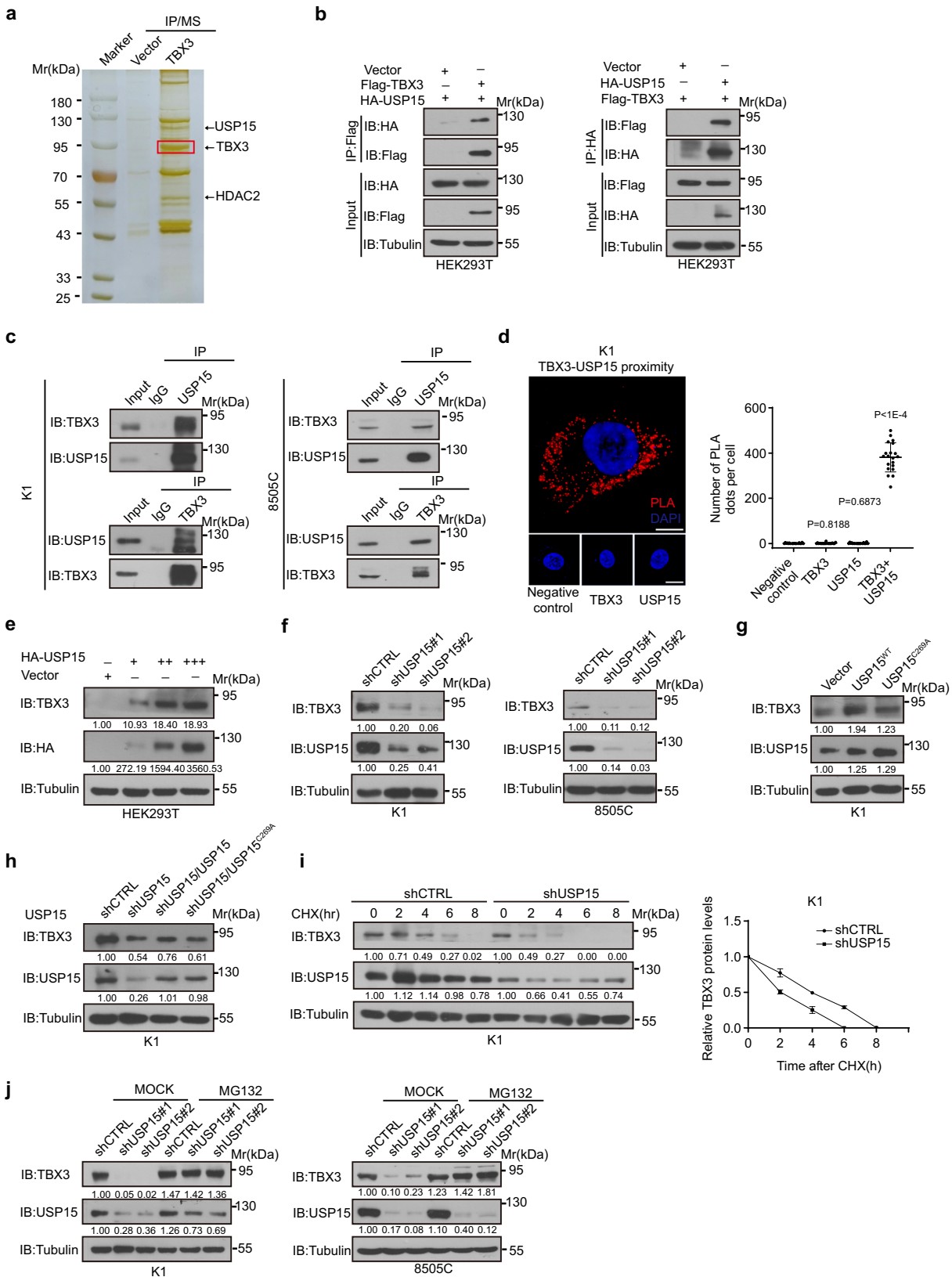

colony formation assays, within K1 and 8505 C, which both represent BRAF$^{V600E}$-mutated thyroid cancer cells [K1 represents PTC cells, 8505 C represents ATC cells] (Fig. 3a, b, Supplementary Fig. 2d–f). While in tumor cells driven by different oncogenic events, such as MCF7 breast cancer cells, the expression and function regulation of USP15 on TBX3 has not shown such direct correlation. Importantly,

reintroducing TBX3 expression efficiently rescued the effect of USP15 knockdown within the above BRAF$^{V600E}$-positive thyroid tumor cells, consistent with the speculation that USP15 acts via TBX3 (Fig. 3a, b, Supplementary Fig. 2d–f). In vivo, USP15 depletion indeed inhibited tumor growth, accompanied with elevated p21$^{CIP1}$, p57$^{KIP2}$, direct TBX3 repressing targets, thus restricted cell proliferation (Fig. 3c–e,

**Fig. 1 | USP15 binds and maintains TBX3 stability. a** Affinity purification and mass spectrometry (IP-MS) was conducted on HEK293T cells over-expressing Flag-tagged TBX3 to analyze TBX3-associated protein complexes. **b** Co-Immunoprecipitations (Co-IPs) were performed to examine the interaction between over-expressed HA-USP15 and Flag-TBX3 in HEK293T cells. **c** Immunoprecipitations (IP) Co-IPs were performed to validate endogenous binding of TBX3 and USP15 in K1 and 8505 C cells. **d** Proximity ligation assay (PLA) was performed to validate colocalization of USP15 and TBX3 in K1 cells. Scale bars, 20 μm. For each sample, at least 20 cells were counted. **e** TBX3 expression was detected after over-expression of different doses of HA-tagged USP15 in HEK293T cells. **f** TBX3 protein expression was detected by IB after USP15 knock-down in K1 or 8505 C cells. **g** TBX3 expression was examined after USP15$^{WT}$ or USP15$^{C269A}$ over-expression in K1 cells. **h** TBX3 expression was detected after USP15 knock-down in K1 cells with or without wild-type (USP15$^{WT}$) or mutant USP15 (USP15$^{C269A}$) re-expression. **i** Half-lives of TBX3 were analyzed after USP15 knock-down in K1 cells. **j** TBX3 expression was examined in USP15 knocked-down K1 or 8505 C cells with or without 20 μM MG132 treatment for 8 h. $n = 3$ biological independent samples (**b**–**j**). Densitometric analyses of western blot were shown (**e**–**j**). Data are shown as the mean ± s.d (**d**, **i**). $P$ values were calculated by unpaired two-tailed Student's $t$ test (**d**). The statistical test used was two-sided (**d**). Uncropped immunoblots and statistical source data are provided in Source Data.

Supplementary Fig. 2g). Remarkably, ectopic TBX3 reversed the function and restored tumor growth (Fig. 3c–e, Supplementary Fig. 2g). Thus, USP15 promotes tumor cell proliferation and tumor growth through maintaining TBX3 level and related downstream events.

To understand how the proteostasis regulation fits in the in vivo highly spontaneous BRAF$^{V600E}$-induced tumorigenesis where Tbx3 is re-activated and critically required for tumor initiation and progression[15], we first collected tumor tissues from mPTC model generated by crossing thyroid peroxidase *TPO-Cre* with *LSL-Braf$^{V600E}$*(Supplementary Fig. 2h)[43]. Interestingly, we detected continuously increased Usp15 expression during the progression of mPTC in parallel with Tbx3, suggesting they could be functionally coupled during tumorigenesis (Fig. 3f and Supplementary Fig. 2i). Therefore, we deleted *Usp15* gene by crossing *Usp15$^{null}$* mice with mPTC line (homozygotes of Usp15 deletion would be referred as mPTC/Usp15$^{-/-}$, heterozygotes as mPTC/Usp15$^{+/-}$ in the following text) to determine the biological significance of Usp15 in endogenous tumor formation (Fig. 3g). Compared to the high penetrance of tumor occurrence around 30 days in control mice, local PTC tumors were hardly formed in mPTC/Usp15$^{-/-}$ and mPTC/Usp15$^{+/-}$ mice (Fig. 3h, i). Statistically, Usp15 loss inhibited the initiation and development of BRAF$^{V600E}$-induced PTC dramatically, even not in the dosage-dependent way (Fig. 3i). Histological staining showed that tumors from mutant mice, when formed, exhibited more differentiated characteristics and retained more healthy follicles compared to those from control mice (Fig. 3j). In these tumors, Tbx3 protein level was significantly decreased, while p57$^{KIP2}$ was increased upon Usp15 loss, consistent with the cellular data (Fig. 3j and Supplementary Fig. 2j). Also in line with the earlier findings that Tbx3 promotes MDSCs recruitment and immune-suppressive tumor microenvironment (TME) construction[15], infiltration of Gr-1 positive MDSCs were significantly reduced in mPTC/Usp15$^{-/-}$ tumor tissues (Fig. 3j). Collectively, these results demonstrated that Usp15 participates in BRAF$^{V600E}$-induced tumor development by maintaining Tbx3 abundance under genetic background.

## BRAF/MAPK cascade achieves high TBX3 abundance via USP15-mediated stabilization

Next, we wondered whether the continuously increased Usp15 through mPTC progressing induced by genetic BRAF$^{V600E}$ incorporation was a tumor cell autonomous behavior. Primary mPTC cells were isolated from mPTC; Rosa26-mTmG reporter line[15], and subjected to BRAF inhibitor PLX4032, or ERK1/2 inhibitor SCH772984 treatment. Notably, Usp15 in the primary tumor cells is under BRAF/MAPK pathway control (Fig. 4a). Comparably, pharmacological inhibition of BRAF or ERK1/2 also significantly down-regulated USP15 expressions in BRAF$^{V600E}$-mutated human thyroid cancer cells (Fig. 4b and Supplementary Fig. 3a). MG132 treatment rescued TBX3 protein loss under BRAF or ERK1/2 inhibitor, indicating that USP15 loss caused TBX3 degradation acceleration accounts part of the overall TBX3 reduction (Fig. 4c and Supplementary Fig. 3b). Since repression of downstream effector AP-1 protein directly reduced USP15 mRNA as well as protein levels (Fig. 4a, d and Supplementary Fig. 3b), we hypothesized that MAPK

directs USP15 expression through transcriptional regulation. Indeed, multiple AP-1 binding motifs were located within USP15 promoter, which were transactivated by c-Jun/JunB/c-Fos over-expression (Fig. 4e). Together, USP15, downstream of activated BRAF/MAPK pathway, functions as the proteostatic controller of TBX3 and determines the ultimate abundance.

Remarkably, induction of Usp15 expression by BRAF/MAPK could be instant and dramatic, so that its level was augmented in E16.5 embryonic thyroid from mPTC mice compared to that from wild-type mice (Fig. 4f and Supplementary Fig. 3c). To further monitor the dynamic reprogramming under BRAF$^{V600E}$-induction, we transduced the immortalized human thyroid cell line Nthy-ori 3-1 with BRAF$^{WT}$ or BRAF$^{V600E}$ and established reliable transformation models. Nthy/BRAF$^{V600E}$ cells gain the capability to form thyroid tumors in vivo, with high metastasis potential and ATC-like transcriptome[44]. USP15 and TBX3 were up-regulated in parallel, even more significantly than the BRAF$^{WT}$-overexpressing cells (Fig. 4g and Supplementary Fig. 3d). Therefore, pathologically activated BRAF/MAPK signaling, possibly by BRAF$^{V600E}$, initiates and promotes tumor development through transcriptionally elevating USP15 and protecting TBX3 from degradation, in addition to direct transcriptionally up-regulating TBX3[15].

## USP15 positively correlates with TBX3 expression through diverse BRAF$^{V600E}$ cancers

To further evaluate the clinical significance of USP15 mediated TBX3 homeostasis and understand their correlation in tumorigenesis, we performed immuno-histochemical (IHC) staining to examine the expression of these factors in patient samples. Across the 152 tissue array samples composed mainly of PTC samples with adjacent tissues, as well as a small number of other types of thyroid cancer tissues, expression levels of USP15 and TBX3 are significantly increased and positively correlated with tumor progression[15] (Supplementary Fig. 4a). Statistical analysis indicated that USP15 level positively correlated with that of TBX3 (Fig. 5a, b, $n = 152$). As we have mentioned, BRAF$^{V600E}$ is the prevalent oncogenic mutation and correlates with dedifferentiation and poor prognosis in PTC, furthermore, USP15-TBX3 axis functions as indispensable downstream effectors of activated BRAF/MPAK pathway. Therefore, we asked whether USP15-mediated TBX3 stabilization could be a molecular signature-related event. We took advantage of another cohort of tissue samples consisted with PTC and the adjacent tissues, with accessible BRAF genetic information ($n = 93$), and analyzed via serial sectioning. Overall, USP15 level still positively correlates with TBX3 level (Fig. 5c, d). Quite interestingly, USP15 expression in BRAF$^{V600E}$-positive samples is significantly higher than that in BRAF$^{WT}$ group (Fig. 5e, f), causing 82.1% of USP15$^{high}$ expression in BRAF$^{V600E}$ samples in comparison to 48.6% in BRAF$^{WT}$ samples (Fig. 5g).

Given the biological significance of USP15 within BRAF$^{V600E}$-related tumorigenesis in vitro, in vivo, and in patients, we are curious about whether the UPS-mediated proteostasis of TBX3 could be a general regulatory hinge to orchestrate TBX3 functional dosage downstream of BRAF/MAPK activation in addition to AP-1 mediated transcriptional control[15]. Indeed, the abundant expression of USP15 and TBX3 concurs

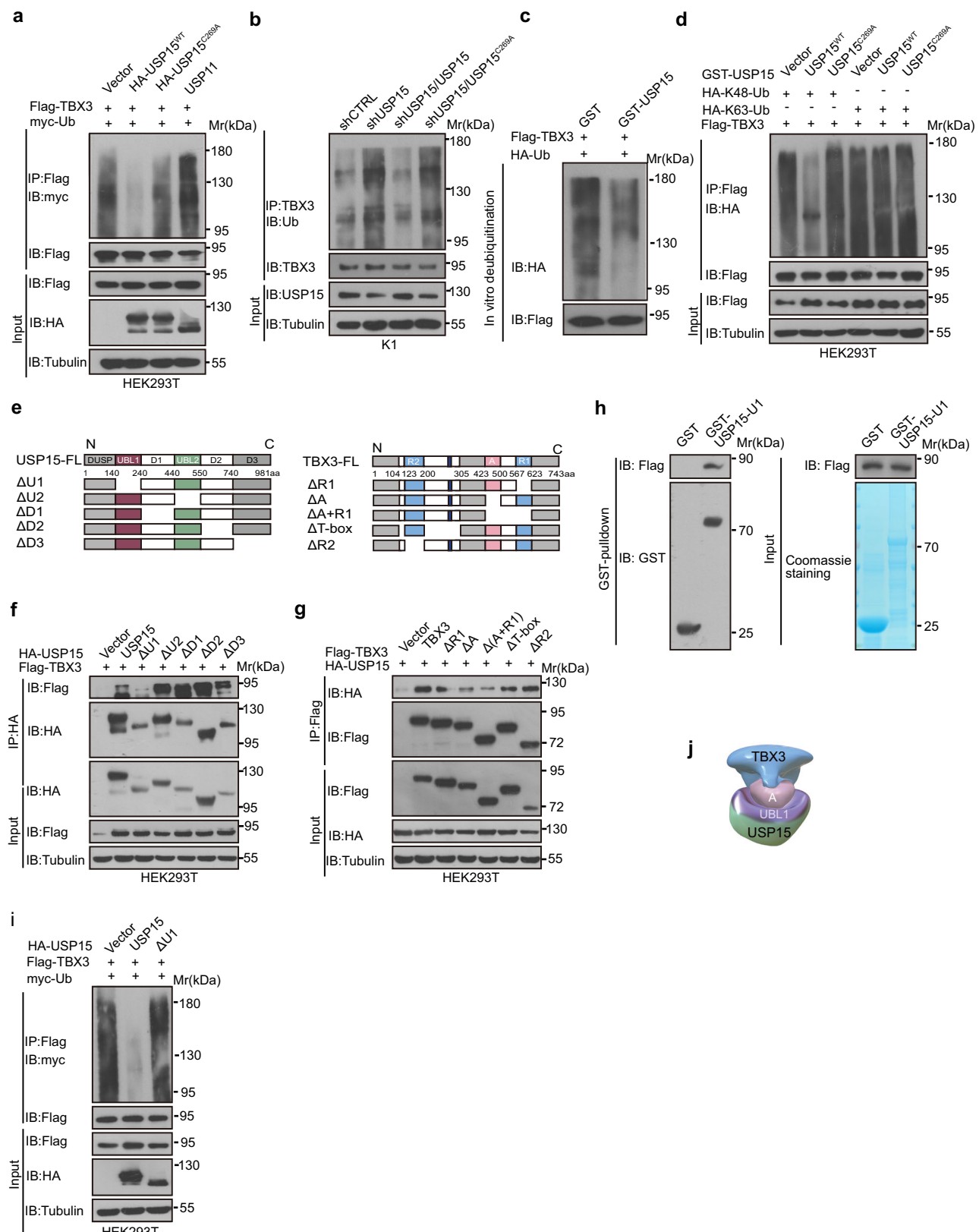

apparently through BRAF^V600E positive cancer cells, such as melanoma cells A2058, OCM-1, A375, besides thyroid cancer cells K1, BCPAP, 8505 C (Supplementary Fig. 4b). Additionally, USP15 and TBX3 also showed high expression within TPC-1 and Cal-62 cells with RET rearrangement or KRAS mutation respectively, indicating the modulation of these proteins may be due to BRAF/MAPK pathway activated with different upstream mutations. Take melanoma as another model where BRAF^V600E-induced BRAF/MAPK activation is also the most dramatic carcinogenic mutation, we found knock-down of USP15 resulted in significant TBX3 reduction (Supplementary Fig. 4c). Consequently, USP15 deficiency inhibits melanoma tumor cell proliferation and xenografted tumor growth in the TBX3-dependent manner, which is

**Fig. 2 | USP15 mediates deubiquitylation of TBX3. a** Ubiquitylation analysis of Flag-tagged-TBX3 after co-transfection with HA-tagged wild-type (USP15[WT]) or mutant USP15 (USP15[C269A]), or USP11 with the presence of myc-tagged-ubiquitin. **b** Ubiquitylation analysis of endogenous TBX3 in USP15 knocked-down K1 cells with or without wild-type (USP15[WT]) or mutant USP15 (USP15[C269A]) re-expression. **c** In vitro deubiquitylation analyses on TBX3 incubated with GST-tagged USP15 as well as HA-tagged-ubiquitin. **d** In vivo deubiquitylation analyses on TBX3 conjugated with indicated ubiquitin chains, also incubated with GST-tagged wild-type (USP15[WT]) or mutant USP15 (USP15[C269A]). **e** Schematic representation of USP15 and TBX3-deletion mutants. **f** Co-IPs were performed to check interaction between different USP15 deletion mutants (ΔU1, ΔU2, ΔD1, ΔD2, ΔD3) and Flag-tagged TBX3. **g** Co-IPs were performed to check interaction between different TBX3 deletion mutants (ΔR1, ΔA, Δ(A + R1), ΔT-box, ΔR2) and HA-tagged USP15. **h** GST pull-down assay was performed to check interaction of purified GST-tagged deletion mutant USP15 (USP15[ΔU1]) and Flag-TBX3 in vitro. Purified GST and GST-USP15[ΔU1] were detected by Coomassie blue staining. **i** Ubiquitylation analysis of Flag-tagged-TBX3 after co-transfection with HA-tagged wild-type or deletion mutant USP15 (USP15[ΔU1]) with the presence of myc-tagged-ubiquitin. **j** Schematic illustration of molecular interaction between USP15 and TBX3. *n* = 3 biological independent samples (**a–d, f –i**). Uncropped immunoblots are provided in Source Data.

consistent with reported USP15 function in melanoma development[42] (Fig. 5h, Supplementary Fig. 4d, e). Similar to the regulatory mechanism under BRAF[V600E]–induced thyroidal transformation, USP15 is also transcriptionally activated by BRAF/MAPK pathway during melanoma genesis and controls TBX3 stability (Fig. 5i, j and Supplementary Fig. 4f). Clinically, TBX3 level positively correlates with that of USP15 as well on our melanoma tissue array (Fig. 5k, l).

Our findings were further supported by available thyroid carcinoma proteomics data from iProX, where USP15 protein level positively correlated with that of TBX3 among samples with both factors detectable[45] (Fig. 5m, *n* = 52). Based on THCA transcriptome data from The Cancer Genome Atlas (TCGA), USP15 expression is positively correlated with TBX3, even BRAF, consistent with the high correlation between TBX3 and BRAF[15] (Supplementary Fig. 4g). Similarly, USP15 positively correlates with BRAF in melanoma (named SKCM) where TBX3 has been confirmed to be highly pro-tumorigenic[46]. USP15 and TBX3 transcriptional expressions also show positively correlated tendency, although not statistically significant (Supplementary Fig. 4g, h). Remarkably, once we sort out BRAF[V600E] mutated patients, the positive correlation between USP15 with TBX3 or with BRAF in both THCA and SKCM was significantly increased (Supplementary Fig. 4h). These analyses indicate that high USP15 expression level could predict higher tumor progression and poorer overall survival, in BRAF[V600E] related malignancies, presumably due to its regulatory function on TBX3 protein homeostasis.

### USP15 coordinates TBX3 abundance through BRAF/MAPK activity-determined development and tumorigenesis

As a developmental factor, Tbx3 is specifically and dynamically expressed during embryonic organogenesis, while its expression goes relatively low in adult organs[9]. BRAF/MAPK pathways are also implicated in physiological cellular processes and organ formation. Specifically, phosphorylated Erks (p-Erk) participate in early embryonic endoderm foregut specification[47], where high p-Erk persists in the descendent organs exemplified with embryonic thyroid (Fig. 6a and Supplementary Fig. 5a). Coincidently, previous microarray combined with laser capture microdissection analysis identifies Tbx3 as a thyroid bud-specific factor at early as E10.5[48]. As embryos develop further, we detected high abundance of Tbx3 at E14.5 when thyroid starting folliculogenesis in parallel with p-Erk (Fig. 6a and Supplementary Fig. 5a).

Together with above findings, we aim to understand whether Usp15 coordinates Tbx3 abundance through physiological and pathological transition, in accordance with BRAF/MAPK pathway. We continuously catalogued expression of these factors across embryogenic, postnatal development, and progressive tumor development, taking the thyroid tissue as a model. Interestingly, Usp15 shows high abundance in parallel with high p-Erk and Tbx3 at E14.5, when thyroid follicles just start differentiation while keeping progenitor characteristics (Fig. 6a and Supplementary Fig. 5a). Their expressions decline as development continues and remain low after birth and in adult thyroid. Remarkably, once postnatal follicle cells initiate transformation under BRAF[V600E]-induction, expression of p-Erk, Tbx3 and Usp15 climbs to the pathological levels (Fig. 6a and Supplementary Fig. 5a).

The finely coordinated expression profile between p-Erk, Tbx3 and Usp15 encouraged us to investigate whether Tbx3 is under the same proteostasis regulation during development. Indeed, Tbx3 was significantly reduced in mouse embryonic fibroblasts (MEFs) derived from *Usp15[null]* mutants (Fig. 6b). We have shown that Tbx3 is restrictedly expressed in the ventral midline of hypothalamus at E10.5[10], which was hardly detectable in the same position during *Usp15[null]* mutant embryogenesis, in spite of comparable mRNA expressions (Fig. 6c and Supplementary Fig. 5b). A series of genetic studies have shown Tbx3 plays fate determination roles for other endoderm-derived organs including the lungs, the liver, and the pancreas[9]. Accordingly, Tbx3 expression within lung mesenchyme at E14.5 was also markedly abolished from *Usp15[null]* mutants[49,50] (Fig. 6c). After birth, Tbx3 protein expression was lost in *Usp15[null]* animal across different tissues, such as the heart, skin, bladder, and thyroid (Fig. 6d). These data suggest that Tbx3 level is subjected to Usp15 regulation from embryonic development, organ formation, and to tumorigenesis, likely responding to BRAF/MAPK cascade.

### Dosage of USP15-TBX3 defines physiological differentiation and pathological de-differentiation

To understand the biological significance between developmentally and pathologically coordinated Usp15 and Tbx3 expression, we compared significance of both factors under each context. Relying on Tbx3 deficient BRAF[V600E]-induced mPTC mice (mPTC/Tbx3[-/-])[15], thyroid developmental models were parallelly obtained by knocking out *Tbx3* with TPO-Cre (Tbx3[Δthy]). Tbx3[Δthy] mutants were born normally, but with disrupted and irregular follicles (Fig. 7a). Gene expression profiles were compared between P90 wild-type and Tbx3[Δthy] thyroid tissues, where most lineage differentiation factors were significantly elevated due to Tbx3 removal (Fig. 7b). Histologically and quantitatively, differentiation factors were increased in Tbx3[Δthy], pointing to an overdifferentiation tendency (Fig. 7a and Supplementary Fig. 5c). The hormone secretion function and T3/T4 levels were thus damaged (Fig. 7c). Given the observation, we were curious about the differentiation state of BRAF[V600E]-driven tumors under Tbx3 deficiency (mPTC/Tbx3[-/-]) (Fig. 7b). Tbx3 knockout restored differentiation factors expression and rendered re-differentiation marked with, but not limited to Tpo, Nis, Tg[15] (Fig. 7a). Taken together, reactivated Tbx3 by BRAF[V600E] plays the same role as in embryonic development, which is to confine the differentiation state.

Strikingly, Usp15 removal dramatically abrogated Tbx3 protein expression from E14.5 thyroid without affecting the mRNA level (Fig. 6c and Supplementary Fig. 5b), thus caused similar developmental defects in terms of disorganized folliculogenesis and elevated differentiation factors, such as Tpo, Tg, and Nis (Fig. 7d). Serum T3/T4 levels were consequently reduced in *Usp15[null]* mutants (Fig. 7e). So BRAF/MAPK directed endoderm development could rely on the highly co-expressed and functionally-coordinated Tbx3 and Usp15 at middle and late embryogenesis (Fig. 7a, d). BRAF[V600E]-induced mPTC was also re-differentiated by Usp15 knockout, largely due to reduced Tbx3 (Fig, 7d, e and Supplementary Fig. 5d).

The next question is whether Usp15-stabilized Tbx3 expression is the determinant for BRAF[V600E]-induced de-differentiation. We first took

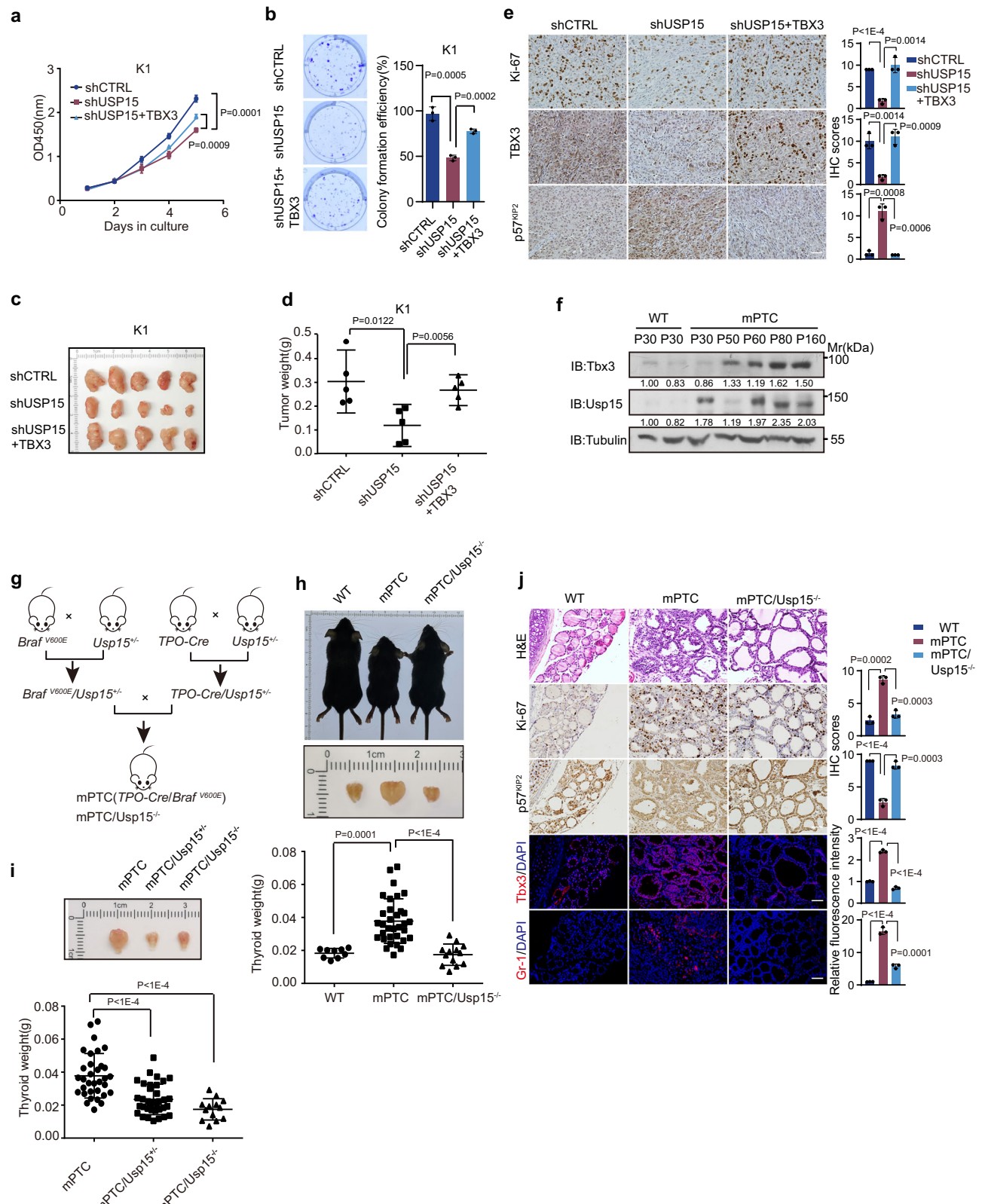

advantage of the Nthy/BRAF^V600E model to monitor the successively pathophysiological transition. Indeed, BRAF^V600E effectively initiates the follicular de-differentiation, represented with dramatically reduced thyroid lineage factors. However, the de-differentiation was reversed by USP15 knockdown and accompanying TBX3 reduction (Fig. 7f and Supplementary Fig. 5e). At the early BRAF^V600E-induced transformation and de-differentiation in vivo, E16,5 mPTC, the initial

proliferation could no longer pursue once Usp15 is absent, with so caused Tbx3 loss (Fig. 7g, Supplementary Fig. 5f, g). Reduced differentiation markers, Tpo and Nis, during tumorigenesis, were also efficiently reacquired and even elevated to normal embryonic follicle levels (Fig. 7g Supplementary Fig. 5f, g). Therefore, USP15 maintains TBX3 expression through physiological development and pathological tumor progression where both confine the differentiation progression,

**Fig. 3 | USP15 mediates BRAF$^{V600E}$-induced tumorigenesis in a TBX3 dependent manner.** USP15 knocked-down K1 cells with or without TBX3 over-expression were subjected to cell growth assays of CCK8 (**a**) or colony formation (**b**). **c** USP15 knocked-down K1 cells with or without TBX3 over-expression were transplanted into athymic mice subcutaneously. **d** Tumor weights from (**c**) were recorded and statistically compared, $n = 5$. **e** IHC stainings of indicated factors were performed on tumor sections from (**c**). Scale bars, 50 μm. **f** IB analysis of Usp15, Tbx3 in thyroid tissues from WT (wild-type) or mPTC (*TPO-Cre; Braf$^{V600E}$*) mice at different time points. **g** The transgenic mice crossing strategy of mPTC and mPTC/Usp15$^{-/-}$ (*TPO-Cre; Braf$^{V600E}$; Usp15$^{-/-}$*) mice. **h** Representative images for whole thyroid tissues from wild-type (WT), mPTC, mPTC/Usp15$^{-/-}$ mice at P30, related thyroid weights were plotted. $n = 9$ WT, $n = 33$ mPTC, $n = 13$ mPTC/Usp15$^{-/-}$. **i** Representative images for whole thyroid tissues from mPTC, mPTC/Usp15$^{+/-}$ and mPTC/Usp15$^{-/-}$ mice at P30, related thyroid weights were plotted. $n = 33$ mPTC, $n = 35$ mPTC/Usp15$^{+/-}$, $n = 13$ mPTC/Usp15$^{-/-}$. **j** Representative H&E, IHC and IF staining on thyroid tissues from WT, mPTC, and mPTC/Usp15$^{-/-}$ littermates. Scale bars, 50 μm. $n = 3$ biological independent samples (**a, b, e, f, j**). Representative images are shown (**b, c, e, h–j**). Data are shown as the mean ± s.d (**a, b, d, e, h–j**). $P$ values were calculated by unpaired two-tailed Student's $t$ test (**a, b, d, e, h–j**). Densitometric analyses of western blot were shown (**f**). Uncropped immunoblots and statistical source data are provided in Source Data.

## Discussion

As a critical factor in embryonic stem cells maintenance, lineage specification, and embryogenic organ formation, TBX3 expression is finely controlled, with its up or down-regulation leading to aberrant regulation of cell fate/proliferation and different malignancies.[13] Prior reports regarding the regulation of TBX3 levels during development and tumorigenesis mainly focus on the transcriptional regulation[9]. Moreover, whether common regulatory mechanisms of TBX3 function through embryogenesis and tumorigenesis was unclear. Our study provides compelling evidence that TBX3 homeostasis is also regulated by the UPS mediator, DUB USP15, during the development and tumorigenesis. The highly conserved TBX3 regulatory mechanism fills the scientific void of TBX3 post-translational modification characterization. Importantly, our study supports TBX3 as a paradigmatic illustration on how reactivation of developmental transcription factors drives carcinogenesis.

Epithelial cell dedifferentiation is recognized as a critical cell state transition during tumorigenesis and confer tumor cells on high proliferation capacity and progenitor-like fate through multidimensional reprogramming[1–3]. Here we observe that the regained differentiation behaviors upon Usp15 or Tbx3 knockout in BRAF$^{V600E}$-induced tumorigenesis, especially the restored thyroid hormone levels, Nis, Tpo, Tg and other differentiation factor expressions, parallel with the elevated differentiation features in Usp15- or Tbx3-knockout organs. Therefore, BRAF/MAPK pathway up-regulates developmental factor TBX3 for tumorigenesis by interfering with the proteostasis module in addition to transcriptional regulation[15], which could be re-activation of the developmental program how BRAF/MAPK intermingles with lineage differentiation factors. Multiple lines of evidences also support the rationale, first of which is the coordinated activation of BRAF/MAPK with Usp15 and stabilized Tbx3 protein during organ development exemplified with the thyroid. Moreover, the high dp-Erk distribution within embryonic foregut presupposes the critical function of BRAF/MAPK in the descendent tissue thyroid, since the liver, another descendent tissue, also needs enough Erks for homeostasis[47,51]. Tissue specific knockout models of Erks will be ideal in the future to further testify the functional correlation between BRAF/MAPK activation and USP15-TBX3 axis in the future. Considering the BRAF/MAPK interaction with multiple developmental factors in tumorigenesis, such as β-catenin, Klf4, Sox2 et al.[10,33–35], it would be interesting to investigate whether conserved regulations also apply.

During embryonic and postnatal development, USP15 is expressed in different tissues, most abundantly in the testes, spleen, thyroid, and adrenal gland. Usp15 global knock-out mice display growth retardation, and the average weights are 20% less compared with wild-type littermates[52]. Here we find Tbx3 is significantly and specifically reduced at the protein level across various tissues in *Usp15$^{null}$* mutants, defining Usp15 as a critical post-translation modifier of Tbx3 through

physiological processes. These findings also raise the question whether the loss of Tbx3, normally expressed in and instructing the development of hypothalamus-pituitary, lung, thyroid, and other organs, is responsible for the developmental defects observed in *Usp15$^{null}$* mutant mice.

Multiple oncogenic pathways have been found to regulate TBX3 expression in occurrence of melanoma, lung, thyroid, breast and other types of cancers[12,15,16,20,21,53,54], here we define USP15 as a critical stabilizer for TBX3. Due to the conflicting functions of substrates, USP15 shows dual effects in tumorigenesis. For instance, USP15 copy number is increased in glioblastoma, breast cancer, and ovarian cancer, but decreased in pancreatic cancer[40,52,55–58]. The current finding provides more evidence about the USP15 oncogenic function, especially during BRAF$^{V600E}$ mutated cancers. Moreover, a few kinases participate in phosphorylation of TBX3, such as AKT3 at S720, cyclin A-CDK2 at either S190 or S354, p38 MAP kinase at S692, and AKT1 is also involved[54,59–61]. Some of the phosphorylation sites also have effect on the stability of TBX3, it will be interesting to check the functional correlation between different types of modifications.

Biochemical interaction partner screening and deubiquitylation analysis supports the USP15-TBX3 axis through different physiological and pathological contexts including PTC development. Xie et al. reported that USP7 promoted PTC progression through directly binding and deubiquitylating TBX3 biochemically in vitro[62]. Even though our findings were validated with a series of in vivo functional assays and generalized across different tumorigenesis and organ formation contexts, the situation that multiple DUBs participate in TBX3 stability regulation cannot be ruled out. Whether USP7 and USP15 deubiquitylate TBX3 at different sites or under different backgrounds needs to be carefully characterized. We and other groups have converged on the critical oncogenic function of TBX3 in various kinds of tumors[9,14,15]. While the transcription factor feature makes TBX3 hard to be targeted, the current findings open up the possibility to modulate TBX3 expression through intervening the enzyme activity of USP15 or the physiological molecular binding. Meanwhile, with the progress of PROTAC technique and definition of specific degradation machinery, it will be possible to accurately modulate the endogenous level of TBX3[63]. Therefore, this study uncovers the dedifferentiation mechanisms under BRAF/MAPK-induced tumorigenesis based on defining the function of USP15 in homeostatic TBX3 regulation during development and tumorigenesis. Translationally, the findings shed light on targeting strategies for USP15-TBX3 axis related developmental and oncogenic diseases and provide more thoughts for BRAF/MAPK related therapy designs.

## Methods

### Ethics approval

For clinical samples, the patients and healthy volunteers all signed an informed consent form issued by the Tianjin Cancer Institute and Hospital Ethics Committee, and the study was approved by the Ethics Committee (Approval number: bc2022062). All mice experiment procedures and protocols were evaluated and authorized by the Regulations of Tianjin Laboratory Animal Management and strictly followed

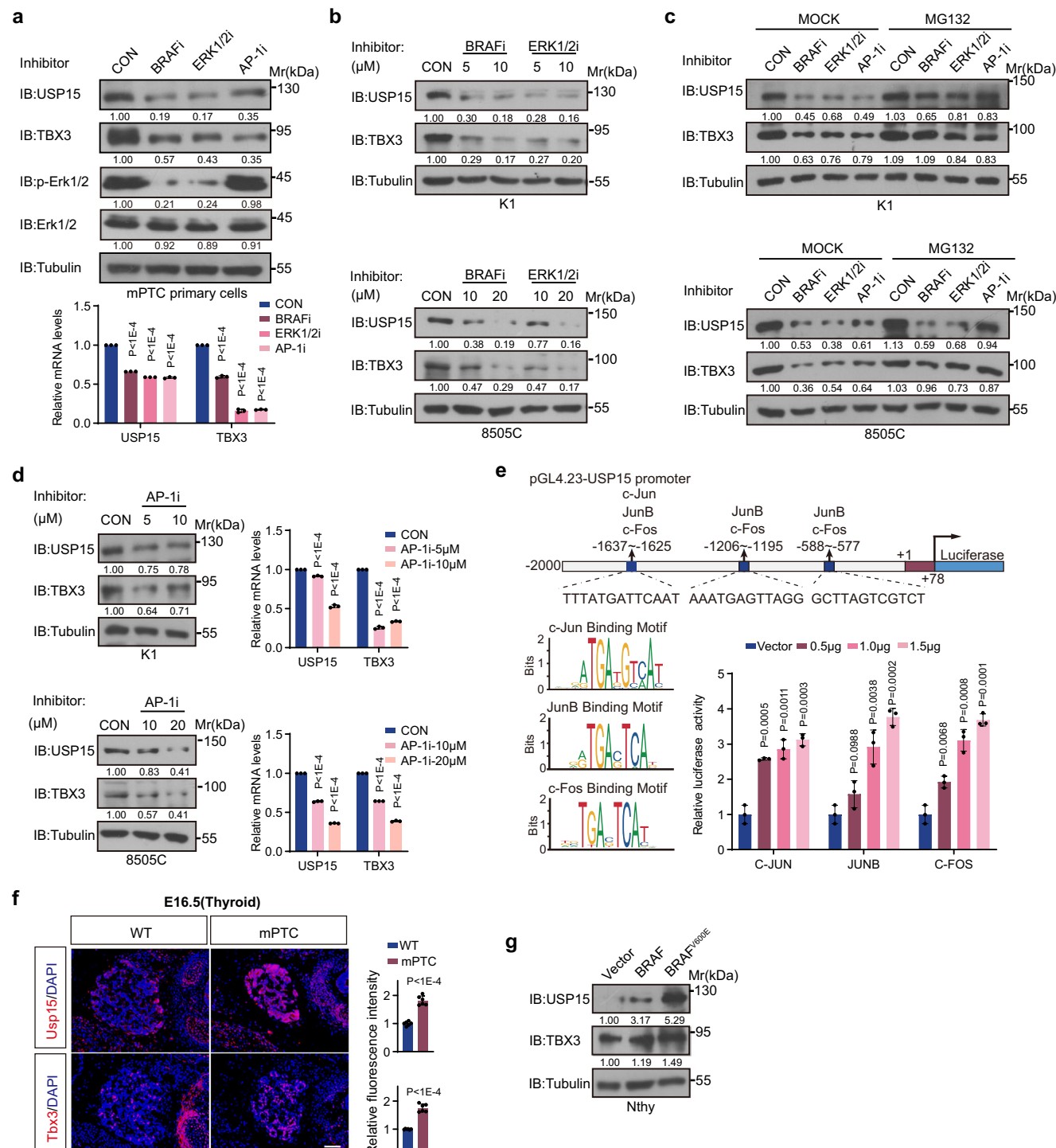

**Fig. 4 | USP15 is transcriptionally upregulated by BRAF/MAPK pathway.**
**a** Usp15, Tbx3, p-Erk1/2, and Erk1/2 protein and mRNA expression was detected by IB or qRT-PCR in mPTC primary cells treated with BRAF inhibitor PLX4032 (10 μM), ERK1/2 inhibitor SCH772984 (10 μM), or AP-1 inhibitor SR11302 (10 μM) for 24 h. **b** USP15 and TBX3 protein expression was detected by IB in K1 or 8505 C cells treated with BRAF inhibitor PLX4032 or ERK1/2 inhibitor SCH772984 for 24 h. **c** USP15 and TBX3 protein expression was detected by IB in K1 or 8505 C cells treated with BRAF inhibitor PLX4032 or ERK1/2 inhibitor SCH772984 for 24 h, followed with MG132 (20 μM) for 8 h. **d** USP15 and TBX3 protein and mRNA expression was detected by IB or qRT-PCR in K1 and 8505 C cells treated with AP-1 inhibitor SR11302 for 24 h. **e** Promoter luciferase activities were analyzed in HEK293T cells

co-transfected with USP15 promoter and different AP-1 factor expression constructs. AP-1 binding motifs were analyzed through USP15 promoter region using Jasper database. **f** IF staining of Usp15 and Tbx3 on transverse sections of E16.5 thyroids from WT and mPTC mice. Scale bars, 50 μm. $n = 6$ WT, $n = 6$ mPTC. Representative images are shown. **g** USP15 and TBX3 protein expression was detected by IB in BRAF[WT] or BRAF[V600E] over-expressed Nthy-ori 3-1 cells. Data are shown as the mean ± s.d (**a**, **d**, **e**, **f**). P values were calculated by unpaired two-tailed Student's t test (**a**, **d**, **e**, **f**). $n = 3$ biological independent samples (**a**–**e**, **g**). Densitometric analyses of western blot were shown (**a**–**d**, **g**). Uncropped immunoblots and statistical source data are provided in Source Data.

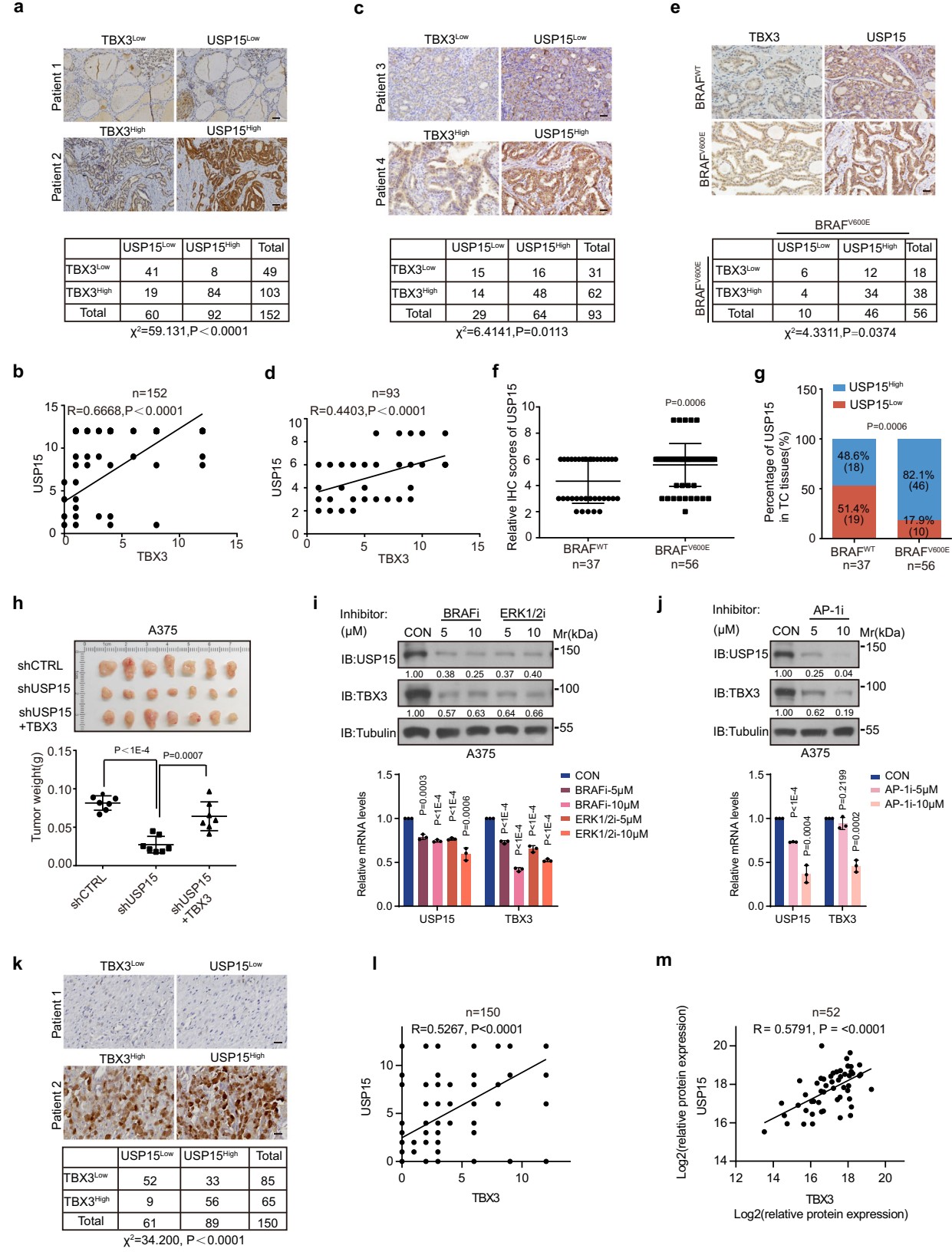

the guidelines under the Institutional Animal Care and Use Committee of Tianjin Medical University (Approval number: SYXK-(Jin) 2019-0004).

## Cell lines

HEK293T cells (ACS-4500) were purchased from the American Type Culture Collection (ATCC). Thyroid cancer cell lines K1, BCPAP, TPC-1, KTC-1, 8505 C, Cal-62, and normal human thyroid cell line Nthy-ori 3-1 (Nthy) were obtained from Tianjin Medical University Cancer Institute and Hospital with STR profiles. Human melanoma cell lines A375, SK-MEL-28, A2058, and mouse melanoma cells B16-F10 were generously provided by Dr Lizhi Hu from Tianjin Medical University. Thyroid cancer cell line KHM-5M (3101HUMSCSP549), human uveal

**Fig. 5 | TBX3 correlates positively with USP15 in BRAF$^{V600E}$-associated tumors.**
**a** Representative images of USP15 and TBX3 IHC stainings on PTC tissue arrays ($n = 152$), included 152 thyroid cancer and adjacent tissues, mostly PTC and with a small amount of other thyroid cancer, correlation analysis was performed by Pearson's chi squared test ($\chi^2$-test), Scale bars, 50 μm. And (**b**) Correlation analysis of USP15 and TBX3 protein levels was performed by Pearson correlation.
**c** Representative images of USP15 and TBX3 IHC stainings on PTC specimens ($n = 93$) independently of the 152-tissue array previously used, correlation analysis was performed by $\chi^2$-test, Scale bars, 50 μm. And (**d**) Correlation analysis of USP15 and TBX3 protein levels was performed by Pearson correlation. **e** Representative images of USP15 and TBX3 IHC stainings on PTC specimens with BRAF$^{WT}$ or BRAF$^{V600E}$ stratification, correlation analysis was performed by $\chi^2$-test, Scale bars, 50 μm. And (**f**) The scores of USP15 staining intensity were plotted, $n = 37$ BRAF$^{WT}$ tissues, $n = 56$ BRAF$^{V600E}$ tissues. **g** Statistical analysis of USP15 expression in BRAF$^{WT}$ or BRAF$^{V600E}$ tissues was performed respectively (for $n = 37$ BRAF$^{WT}$ tissues, USP15 high $n = 18$; for $n = 56$ BRAF$^{V600E}$ tissues, USP15 high $n = 46$) by $\chi^2$-test. **h** A375 cells

with USP15 knock-down with or without TBX3 over-expression were transplanted into athymic mice subcutaneously, Tumor weights were recorded and statistically compared, $n = 7$. **i** USP15 and TBX3 protein and mRNA expression was detected by IB or qRT-PCR in A375 cells treated with BRAF inhibitor PLX4032 or ERK1/2 inhibitor SCH772984 for 24 h. **j** USP15 and TBX3 protein and mRNA expression was detected by IB or qRT-PCR in A375 cells treated with AP-1 inhibitor SR11302 for 24 h.
**k** Representative images of USP15 and TBX3 IHC staining on melanoma tissue arrays ($n = 150$), correlation analysis was performed by $\chi^2$-test, Scale bars, 50 μm. And (**l**) Correlation analysis of USP15 and TBX3 protein levels was peformed by Pearson correlation. **m** Correlation analysis between TBX3 and USP15 based on iProX databases (IPX0001444000, $n = 52$). Data are shown as the mean ± s.d (**f**, **h**, **i**, **j**). $P$ values were calculated by unpaired two-tailed Student's $t$ test (**f**, **h**, **i**, **j**). The statistical test used was two-sided (**a**, **c**, **e**, **g**, **k**). $P$ values (two-tailed) were calculated by Pearson r (**b**, **d**, **l**, **m**). $n = 3$ biological independent samples (**i**, **j**). Densitometric analyses of western blot were shown (**i**, **j**). Uncropped immunoblots and statistical source data are provided in Source Data.

melanoma cell lines OCM-1, MUB-2B, human breast cancer cell line MCF-7 (1101HUM-PUMC000013) and human lung adenocarcinoma cell line A549 (1101HUM-PUMC000002) were obtained from Cell Resource Center, Peking Union Medical College (PCRC). HEK293T, A375, TPC-1, K1, OCM-1, B16-F10 cell lines were maintained in DMEM medium supplemented with 10% fetal bovine serum (FBS). Nthy, 8505 C, BCPAP, MUB-2B, Cal-62, KTC, KHM-5M were maintained in RPMI 1640 medium supplemented with 10% FBS. $Usp15^{+/+}$ and $Usp15^{-/-}$ MEFs cells were isolated from E13.5 embryos of mice and cultured in DMEM supplemented with 10% FBS and 100 μg/mL streptomycin. All the cells were cultured in a humidified incubator equilibrated with 5% CO$_2$ at 37 °C.

## Mice

Animals used in this study were maintained as specific-pathogen free (SPF) mice on a C57BL/6 genetic background, both sexes we used at E10.5–6 months and at least three animals were analyzed for each genotype. The $Usp15$ global knockout mice ($Usp15^{+/-}$) were kindly gifted from Dr Huadong Pei[56]. $TPO$-$Cre$ mice were kindly gifted from Dr Shioko Kimura[64], $LSL$-$Braf^{V600E}CA$ mice were kindly gifted from Dr Martin McMahon[65]. $TPO$-$Cre$ mice were bred to $LSL$-$Braf^{V600E}CA$ mice to generate $TPO$-$Cre$; $LSL$-$Braf^{V600E}$ mice (mPTC model)[43]. To generate $Usp15$ deletion mPTC model, the $TPO$-$Cre$; $Usp15^{+/-}$ mice were crossed with the $LSL$-$Braf^{V600E}CA$; $Usp15^{+/-}$ mice to obtain $TPO$-$Cre$; $LSL$-$Braf^{V600E}$; $Usp15^{-/-}$ mice (mPTC/Usp15$^{-/-}$). Tbx3$^{flox/flox}$ mice were kindly gifted from Dr Anne Moon. $Rosa26$-$mTmG$ mice were purchased from The Jackson Laboratory. $TPO$-$cre$ mice were bred to $Rosa26$-$mTmG$ to generate $TPO$-$cre$; $Rosa26$-$mTmG$ mice, $LSL$-$Braf^{V600E}CA$ mice were crossed with the $TPO$-$cre$; $Rosa26$-$mTmG$ mice to generate $TPO$-$cre$; $Braf^{V600E}$; $Rosa26$-$mTmG$ mice. Thyroid tumors were collected at different time points to analyze the development of tumors. Mouse embryos were obtained from 8 to 16-week-old female mice, mated with male mice. Female and male mice were naturally mated and kept together, insemination was verified the next morning by the presence of a copulatory plug, and this day was defined as embryonic day 0.5 (E0.5).

## Plasmids

The plasmid constructs expressing HA-USP15, Flag-TBX3, were cloned into lentiviral vector pCDH-CMV-puro. Flag-tagged full length and truncated mutants of TBX3 were cloned into lentiviral vector pCDH-CMV-puro as well. The HA-tagged full length and truncated mutants of USP15 (U1, U2, D1, D2, D3) were kindly gifted from Dr Huadong Pei. Full length (2078 bp) for USP15 promoter was cloned into pGL4.23 vector. Recombinant lentiviruses expressing different shRNAs were obtained by cloning designed shRNA into pLKO.1-puro. shRNA sequences are listed in Supplementary Data 2.

## Antibodies and reagents

Antibodies used for immunoblot (IB), immunoprecipitation (IP), immunohistochemistry (IHC) and immunofluorescence (IF) were as follows. Anti-TBX3 antibodies for IB Anti-TBX3 (#ab99302, Abcam, 1:500). Anti-Flag (#3165, 1:1000), Anti-α-Tubulin (T5168, 1:15000) was purchased from Sigma. Anti-HA (#3724, 1:1500), Anti-Ubiquitin (#3936, 1:500), Anti-GST (#2622, 1:1000), was purchased from Cell Signaling Technology. Anti-USP15 (#A300-923A, BETHYL, 1:1000), Anti-USP15 (#67557-1-Ig, Proteintech 1:200), Anti-Ki-67 (#ab15580, Abcam, 1:100), Anti-Tpo (#ab278525, Abcam, 1:200), Anti-Tg (#ab156008, Abcam, 1:200), Anti-Nis (#MA5-12308, Thermo, 1:400), Anti-Gr-1 (#108401, BioLegend, 1:200), Anti-PAX8 (#ab53490, Abcam, 1:10), Anti-TTF1 (#ab76013, Abcam, 1:200) were employed according to the instructions. Reagents used in this study included MG132 (#S2619, Selleck), Puromycin (#P8230, Solario), CHX (#HY-12320, MCE), Blasticidin (#B9300, Solario), PLX4032 (#S1267, Selleck), SCH772984 (#S7101, Selleck), SR11302 (#E2813, Selleck).

## Mass spectrometry analysis

Flag-tagged TBX3 was transfected into HEK293T cells for 48 h, lysed with lysis buffer (50 mM Tris-HCl, pH7.4, 150 mM NaCl, 5 mM EDTA, 0.5% NP-40) freshly supplemented with protease inhibitor cocktail. Whole cell lysates were then incubated with Flag-M2 beads to precipitate Flag-tagged TBX3 and associated partners. Beads were washed five times using the lysis buffer, then eluted with 100 μg/mL FLAG peptides (Sigma-Aldrich) in cold PBS. The eluted proteins were resolved in 5 × SDS loading buffer, boiled at 99 °C for 10 min. The eluted proteins were separated by SDS-PAGE Gel, silver stained using Pierce silver staining Kit (Invitrogen), and then in-gel tryptic digestion, subjected to Thermo Scientific TM Q Exactive$^{TM}$ Plus. $n = 2$ biological replicates. Secondary mass spectrometry data were retrieved using Proteome Discoverer 1.3.

## Immunoprecipitation

For immunoprecipitation, 60 μL of 50% protein A/G Agarose beads (Invitrogen) were incubated with control or specific antibodies (2–4 μg) for 8 h at 4 °C with constant rotating, then centrifuged at $500 \times g$ for 5 min at 4 °C. Whole cell lysates were re-suspended by prepared cold lysis buffer (50 mM Tris-HCl, pH = 7.4, 150 mM NaCl, 5 mM EDTA, 0.5% NP-40 supplemented with fresh protease inhibitors cocktail) for 1 h at 4 °C, centrifuged at $13,400 \times g$ for 15 min at 4 °C. 500 μg of protein was then added and the incubation was continued for overnight at 4 °C. Beads were then washed five times using the lysis buffer, eluted in 2 × SDS loading buffer and boiling for 10 min. The boiled sample complexes were subjected to SDS-PAGE followed by immunoblotting with appropriate antibodies.

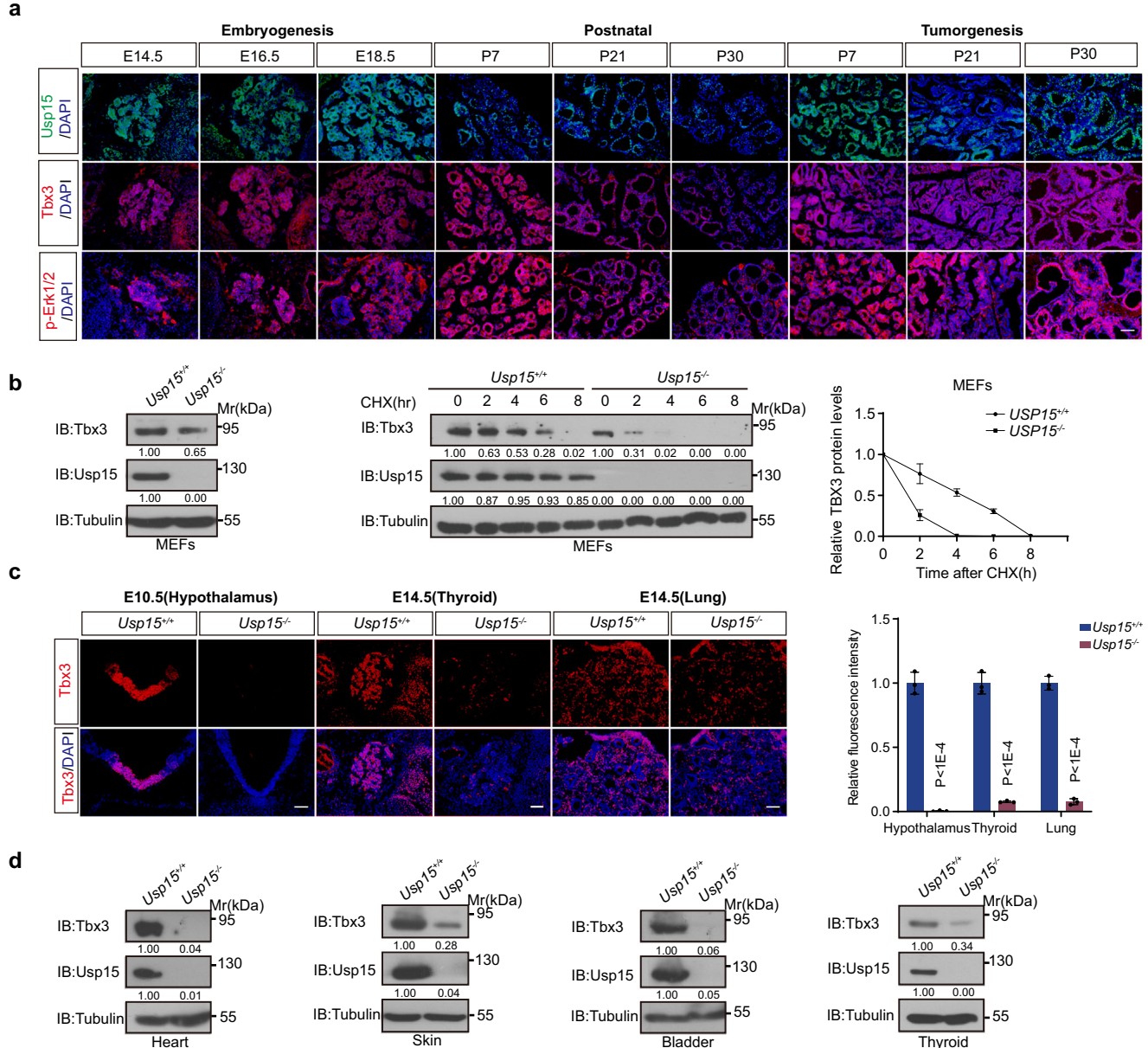

**Fig. 6 | USP15 regulates TBX3 abundance throughout developmental and pathological processes. a** IF staining of Usp15, Tbx3 and p-Erk1/2 during thyroid development and tumorigenesis (*n* = 3). Scale bars, 50 μm. **b** IB expression of Usp15 and Tbx3, and Tbx3 half-life was analyzed in MEFs from wild-type (*Usp15*⁺/⁺) or *Usp15*^null^ (*Usp15*⁻/⁻) mutant mice. (*n* = 3). **c** IF staining of Tbx3 on sections of E10.5 hypothalamus, E14.5 thyroid and lung from wild-type (*Usp15*⁺/⁺) or *Usp15*^null^ mutant mice (*n* = 3). Scale bars, 50 μm. The right panel showed relative fluorescence

intensities. **d** IB analysis of Usp15 and Tbx3 in different adult tissues from wild-type (*Usp15*⁺/⁺) or *Usp15*^null^ mutant mice. (*n* = 3). *n* = 3 biological independent samples (**a**–**d**). Data are shown as the mean ± s.d (**b**, **c**). *P* values were calculated by unpaired two-tailed Student's *t* test (**c**). Densitometric analyses of western blot were shown (**b**, **d**). Uncropped immunoblots and statistical source data are provided in Source Data.

## Protein half-life assay

For TBX3 half-life assay, cancer cell lines stably infected with indicated lentiviruses, or HEK293T cells transfected with indicated constructs were treated with the protein synthesis inhibitor cycloheximide (CHX, 100 μg/mL) for indicated time before harvesting. Immunoblotting was carried out to detect TBX3 protein level.

## In vivo deubiquitylation assay

Cells transfected by indicated constructs for 48 h were subjected to in vivo ubiquitylation assays. After treated with the proteasome inhibitor MG132 (20 μM; Selleck) for 8 h, cells were collected in lysis buffer, then the lysates were pre-cleared by centrifugation at 13,400 × *g* for

15 min. Whole cell lysates (500 μg) was then incubated with Flag-M2 beads overnight at 4 °C. After five times washes, western blot was carried out to detect ubiquitylated TBX3 with anti-HA (or anti-Ub) monoclonal antibody.

## In vitro deubiquitylation assay

GST-fusion deubiquitylation enzyme USP15 protein were purified from *BL21* strain, binding with GST-4B sepharose and eluted with GSH (1×Reduced glutathione) buffer. Flag-tagged TBX3 and HA-tagged ubiquitin were co-transfected into HEK293T cells for 48 h, and then treated with 20 μM MG132 for 8 h. HEK293T cells were then lysed with denaturing buffer (10 mM Tris-HCl, pH 8.0; 100 mM NaCl; 1 mM EDTA;

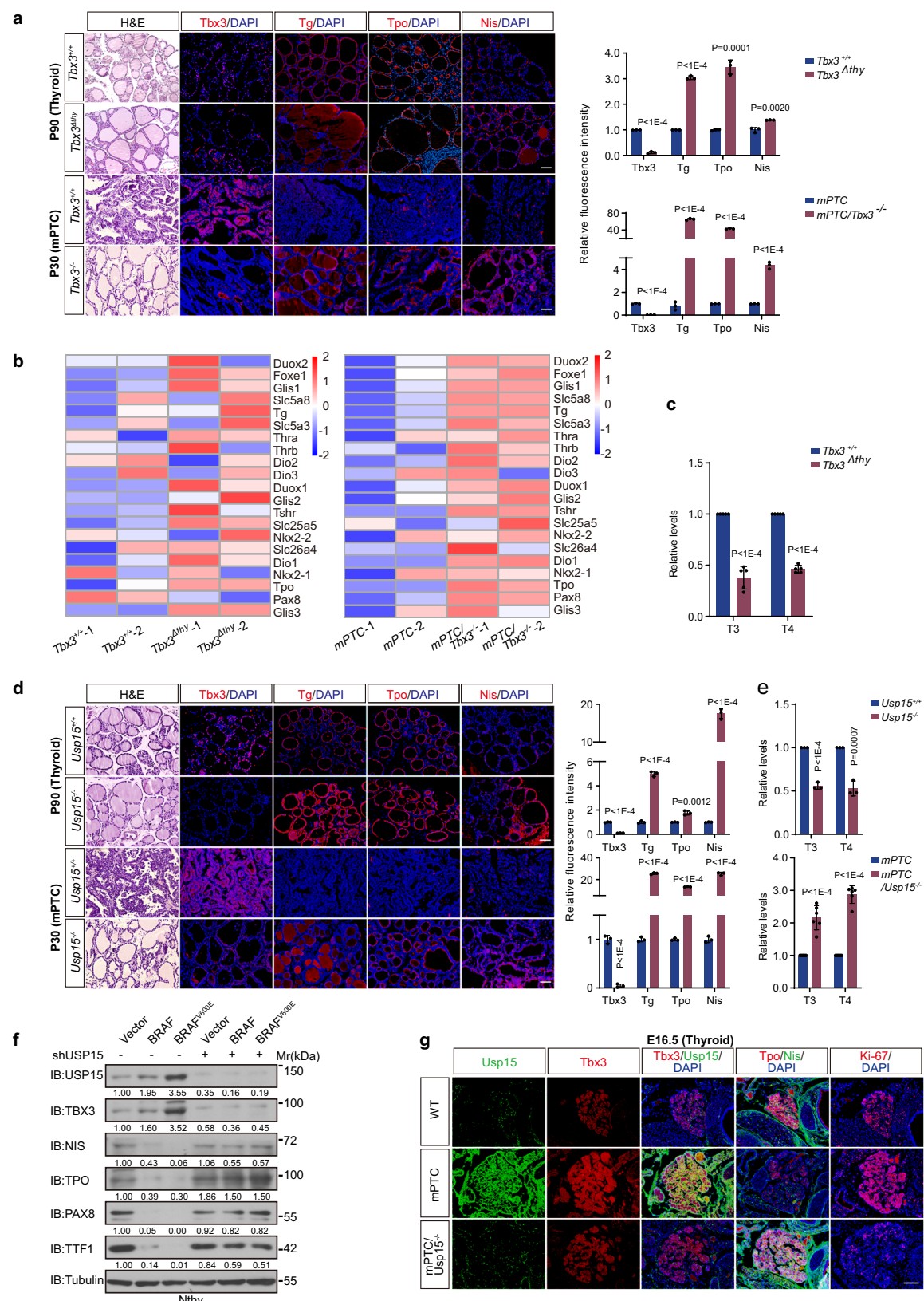

0.5% NP-40; 2% SDS; 20 mM NEM) freshly supplemented with protease inhibitor cocktail. The lysates were boiled for 10 min at 95 °C, diluted in NETN buffer and incubated with HA-agarose beads, then eluted with HA-peptide, and collecting elutions binding with Flag-M2 beads were collected. Purified GST-fusion USP15 protein was then add into the same amounts of beads rotated in the reaction buffer (50 mM Tris-HCl, pH = 8.0; 50 mM NaCl, 1 mM EDTA; 10 mM DTT; 5% glycerol) for 8 h at

37 °C, and then the samples were eluted with 2×SDS loading buffer. Western Blot was carried out to detect ubiquitylated TBX3 with anti-HA monoclonal antibody.

**GST pulldown assay**
For GST pulldown assay, 2 μg GST-USP15-U1 protein were incubated with 10 μg of Flag-TBX3 protein in binding buffer 50 mM Tris-HCl 7.5,

**Fig. 7 | USP15 and TBX3 confines differentiation progression comparably through BRAF/MAPK activity directed embryogenesis and tumorigenesis.** **a** H&E or IF staining of Tbx3, Tg, Tpo and Nis on transverse sections of P90 thyroid from wild-type (*Tbx3*$^{+/+}$) or Tbx3 thyroid specific knock-out mutant (*Tbx3*$^{\Delta thy}$), or mPTC and mPTC/Tbx3$^{-/-}$ mice (*n* = 3). Scale bars, 50 μm. The right panels showed relative fluorescence intensities. **b** Heatmap representation of thyroid differentiation (TD) genes in thyroid tissues from *Tbx3*$^{+/+}$, *Tbx3*$^{\Delta thy}$, mPTC and mPTC/Tbx3$^{-/-}$ mice. **c** The relative levels of thyroid hormones T3 and T4 from *Tbx3*$^{+/+}$, *Tbx3*$^{\Delta thy}$ mice (*n* = 5) were detected. **d** H&E or IF staining of Tbx3, Tg, Tpo and Nis on transverse sections of P90 thyroid from wild-type (*Usp15*$^{+/+}$) or Usp15$^{null}$ mutant mice (*Usp15*$^{-/-}$), as well as P30 mPTC mice (*n* = 3). Scale bars, 50 μm. The right panels

showed relative fluorescence intensities. **e** The relative T3, T4 levels in *Usp15*$^{+/+}$ and *Usp15*$^{-/-}$, or mPTC (*n* = 3) and mPTC/*Usp15*$^{-/-}$ mice (*n* = 6). **f** The indicated proteins expression was detected after USP15 knock-down in Nthy-ori 3-1 cells with or without wild-type (BRAF$^{WT}$), BRAF (BRAF$^{V600E}$) overexpression. **g** IF staining of Tbx3, Tg, Tpo, Nis and Ki-67 on transverse sections of thyroid tissues from wild-type or mPTC and mPTC/*Usp15*$^{-/-}$ E16.5 mice (*n* = 3). Scale bars, 50 μm. *n* = 3 biological independent samples (**a**, **d**, **f**, **g**). Representative images are shown (**a**, **d**, **g**). Data are shown as the mean ± s.d (**a**, **c**, **d**, **e**). *P* values were calculated by unpaired two-tailed Student's *t* test (**a**, **c**, **d**, **e**, **g**). Densitometric analyses of western blot were shown (**f**). Uncropped immunoblots and statistical source data are provided in Source Data.

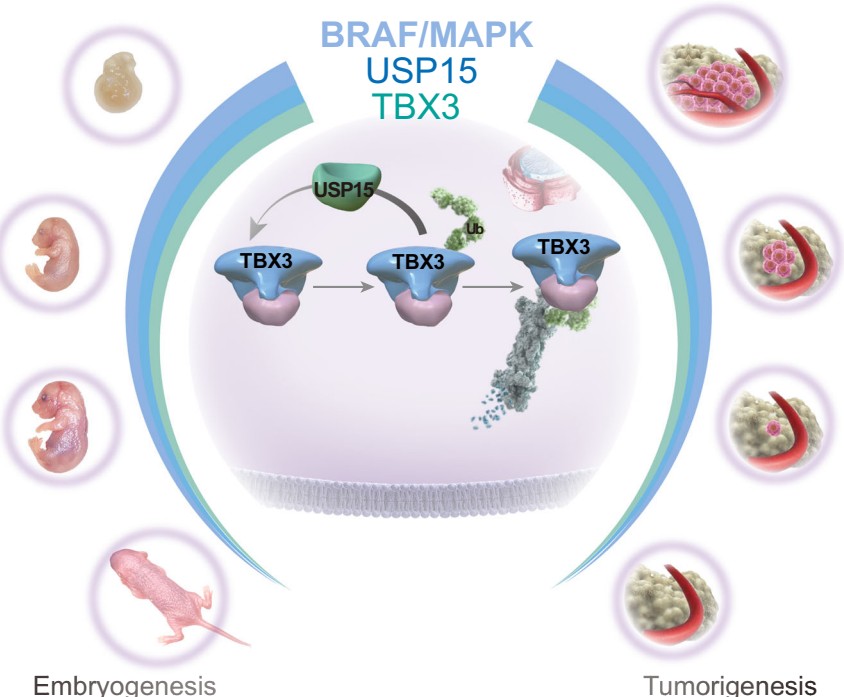

Embryogenesis　　　　　　　　　　　　　　　　Tumorigenesis

**Fig. 8 | Scheme depicting the molecular mechanisms underlying TBX3 proteostasis regulation through development and tumorigenesis downstream of BRAF/MAPK.** USP15 was identified as a DUB, who regulates TBX3 abundance during BRAF/MAPK-directed lineage differentiation, epithelial transformation, and cancer progression. At de-differentiating cell state, early embryonic and advanced oncogenic stages, high BRAF/MAPK/USP15 secures high abundance of TBX3. While at differentiating cell state, postnatal tissue and initial transformation, low BRAF/MAPK/USP15 depletes TBX3 respectively.

---

500 mM NaCl, 1% NP-40, 0.5 mM EDTA, 1 mM PMSF plus protease inhibitors (Roche) at 4 °C overnight, followed by an additional 1 h Glutathione Sepharose beads incubation. The beads were then washed six times and analyzed using SDS-PAGE and western blotting.

### Immunohistochemistry, immunofluorescence and PLA
Tissues were dissected, fixed in 4% paraformaldehyde (PFA) overnight. For immunohistochemistry, samples were embedded in paraffin and sectioned at 5 μm. Primarily sections were deparaffinized, rehydrated and subjected to antigen retrieval in either sodium citrate, pH = 6.0 (Ki-67) or Tris-EDTA pH = 8.0 (USP15, TBX3). After dislodging endogenous peroxidase with 3% H$_2$O$_2$, sections were permeabilized in 0.1% TritonX-100 for 30 min, and blocked in 10% goat serum for 1 h. Sections were incubated overnight at 4 °C in primary antibodies, washed in PBS-T, developed using DAB Substrate Kit (ZSGB BIO, China) and counterstained with Haematoxylin. For immunofluorescence, tissues were embedded in OCT and cell lines fixed in 4% paraformaldehyde for 15 min. Samples as described above were incubated in primary antibodies, followed by incubation for 1 h at room temperature with secondary antibodies conjugated with different fluorochromes. Nuclei were counterstained with DAPI. Sections images were used for confocal detection.

For the PLA assays (Duolink), we followed the manufacturer's instructions (Sigma, DUO92102) with minor modifications. In brief, cells were seeded in glass bottom cell culture dish. After 24 h, cells were washed with PBS and fixed with 4% PFA for 15 min at room temperature. Cells were then sequentially washed twice with PBS, permeabilized with 0.5% Triton X-100 for 10 min at room temperature, washed three times with PBS, blocked, incubated with antibodies at 4 °C overnight and processed according to the manufacturer's instructions. Images were acquired with a Leica THUNDER Imager system. Red spots indicate protein–protein interactions, and blue indicates DAPI-stained nuclei.

### Cell proliferation and colony formation assay
For cell proliferation assay, K1, 8505 C or A375 cells stably infected with USP15 shRNA or TBX3 overexpression lentiviruses were plated at 2000 cells per well in 96-well plates and cultured overnight. The cell proliferation index was measured using a Cell Counting Kit-8 (NCM Biotech, China) at 0, 24, 48, 72 and 96 h post-plated according to the manufacturer's instruction. For colony formation assay, cells were seeded into 6-well plates at a density of 300–500 cells per well and cultured for 2 weeks. After fixed with methanol, the cells were stained

with 1% crystal violet 30 min and then the colonies counted and representative areas were imaged.

## Tissue arrays

The human PTC tissue arrays were purchased from Shanghai Outdo Biotech company included 152 thyroid cancer and adjacent tissues, mostly PTC and with a small amount of other thyroid cancer. A total of 93 cases of pathological sections were obtained in Tianjin Medical University Cancer Institute and Hospital to distinguish whether BRAF is mutated or not, including 37 cases of BRAF$^{WT}$ and 56 cases of BRAF$^{V600E}$ mutation. Melanoma tissue arrays were provided by Dr Jilong Yang from Tianjin Medical University Cancer Institute and Hospital. USP15, TBX3 protein expression levels of all the samples were scored as four grades (negative, +, ++, +++) by multiplying the percentage of positive cells and immunostaining intensity. The percentage was scored as following: non-positive cells as 0 point, 1–30% as 1 point, 31–60% as 2 points, 61–80% and 81–100% as 3 and 4 points, respectively. The intensity of staining was scored: no positive staining as 0 point, weak staining as 1 point, moderate staining as 2 points, strong staining as 3 points. The final scores were obtained according to above terms: 0 point was no expression, 1–3 was low expression, 4–8 was moderate expression, and 9–12 was high expression.

## In situ hybridization

Embryos from 10.5 or 14.5 of *Usp15*$^{+/+}$ and *Usp15*$^{-/-}$ gestational mice were frozen in optimal cutting temperature compound (SAKURA) at −80 °C, and sectioned at a thickness of 10 μm. TBX3 in situ hybridizations were performed according to previous protocol[10,66]. Digoxigenin-UTP labeling and in vitro transcription of plasmid cDNA into antisense were performed using the DIG RNA Labeling Kit (Roche). Hybridization was performed in 150 μL hybridization solution [10 mM Tris-HCl (pH = 7.5), 600 mM NaCl, 1 mM EDTA (pH = 7.5), 0.25% SDS, 10% Dextran Sulfate, 1 × Denhardts (Thermo), 200 μg/mL yeast tRNA (Thermo), 50% formamide] with 800 ng TBX3 probe per slide at 65 °C overnight. Slides were blocked in 20% sheep serum and 2% BMB for 1 h and then incubated in anti-Digoxigenin-AP antibody (Roche, 1:2500) with 5% sheep serum and 2% BMB at 4 °C overnight. In situ signals were detected using BM purple substrate (Roche).

## Tumor xenografts

Six-eight-week-old female BALB/c nude mice were purchased from Beijing Vital River Laboratory Animal Technology Company (Vital River, Beijing, China). 1 × 10$^6$ K1 or A375 cells stably infected with USP15 shRNA or TBX3 overexpression lentiviruses were injected subcutaneously into the dorsal flanks of the mice ($n = 5/7$). Tumor volumes were measured approximately every 3 days and calculated according to the following formula: Volume (mm$^3$) = 1/2 × length × width$^2$. Mice were euthanized while the smallest tumor was detectable and the largest tumor was strictly under 2000 mm$^3$ according to the ethics committee requirements. Paraffin embedding and immunohistochemical analysis were then carried out. All procedures were approved by the Animal Care Committee of Tianjin Medical University.

## Primary cell culture

Cancer tissues obtained from *TPO-cre; Braf*$^{V600E}$*; Rosa26-mTmG* were minced in lysis buffer (Collagenase I (Sigma), 1 mg/mL; Dispase II (Invitrogen) 0.5 mg/mL in PBS) and shaked on the orbital shaker at 37 °C for 60 min. Dissociated tumor cells were filtered with 40 μm cell strainer (#352340, BD Falcon) and centrifuged at 500 g for 5 min, then the pellet was resuspended in cold Ham's F-12 Nutrient Mix (#11765062, Gibco). The isolated tumor cells were washed with PBS and then cultured in F-12 medium supplemented with 10% FBS, 1% penicillin/streptomycin, 5 mg/L transferrin (Sigma-Aldrich), 10 mg/L bovine insulin (Solarbio), 3.5 mg/L hydrocortisone (Sigma-Aldrich), 10 mg/L somatostatin (Sigma-Aldrich), 0.02 mg/L Gly-His-Lysacetate

(Sigma-Aldrich) and 1 IU/L bovine thyroid stimulating hormone (TSH) (Sigma-Aldrich) related to the 6H medium at 37 °C. Next, Primary cells treated with 10 μM BRAFi (PLX4032), 10 μM ERKi (SCH772984) or AP-1i (T-5224) were subjected to western blot and RT-qPCR analysis.

## Luciferase reporter construct and reporter activity assay

Full length (2078 bps) USP15 promoter was cloned into pGL4.23 vector. HEK293T were seeded in 24-well plates and then co-transfected with different doses of AP-1 expression plasmids and pGL4.23-USP15 promoter after 48 h. Luciferase activities were determined with the Dual-Luciferase® Reporter Assay System (Promega, USA). Primer sequences are listed in Supplementary Data 2.

## RT-qPCR

A quantitative real-time RT-PCR method was used to measure the amount of RNA. Briefly, total RNA was extracted using TRIzol (Invitrogen). Reverse transcription of RNA was performed using the RevertAid First Strand cDNA Synthesis Kit (Thermo Fisher Scientific) and qPCR was performed using SYBR Green Master Mix (Vazyme) according to the manufacturer's instructions. Primer sequences are listed in Supplementary Data 2.

## RNA-Seq

TRIzol (Invitrogen) was used to isolate total RNA from primary normal thyroid. A total of RNA samples were submitted for sequencing to the Annoroad Gene Company (Beijing, China) and MGI DNBSEQ T7. The sequence reads were mapped to mouse reference genome (mm10). HTSeq 0.6.0 and DESeq2 1.20.0 were used to analyzed results with cutoff ($p < 0.05$, fold change >1.5).

## Proteomics data analysis

Proteomics data for the study were obtained from the iProX databases (https://ngdc.cncb.ac.cn/, IPX0001444000), and then we analyzed the correlation of USP15 and TBX3 in co-expressed normal thyroid or tumor tissues ($n = 52$)[45]. GraphPad Prism 9 software was used to generate the Correlational analysis and to calculate the P-value.

## Statistical analysis

All results are shown as the mean ± s.d. of multiple independent experiments. All statistical analyses were performed with GraphPad Prism 9 software. All statistical tests were two-sided, and $P < 0.05$ were considered to be statistically significant.

## Reporting summary

Further information on research design is available in the Nature Portfolio Reporting Summary linked to this article.

## Data availability

IP-MS raw data for HEK293T cells were deposited in iProX (IPX000821100). The RNA-seq data generated in this study has been deposited in the Gene Expression Omnibus (GEO) under the accession GSE255930 available from GEO datasets. GEO Accession Number: GSE166513, TCGA-THCA dataset (https://portal.gdc.cancer.gov/projects/TCGA-THCA) and TCGA-SKCM dataset (https://portal.gdc.cancer.gov/projects/TCGA-SKCM) were used for analyses in this study. The remaining data are available within the Article, Supplementary Information or Source Data file. Source data are provided with this paper.

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

## Acknowledgements

The authors would like to thank Dr Huadong Pei for kindly providing the Usp15null genetics mouse models. We also thank Dr Martin McMahon for kindly sharing the *Braf*CA (*LSL-BRAF*V600E) mouse model, Dr Shioko Kimura for sharing the *TPO-Cre* mouse model, Dr Anne Moon for kindly providing the Tbx3flox/flox genetic mouse models. This work was supported in part by the National Natural Science Foundation of China (82273078 and 82073052 to L.Z., 82372600 to H.G., 32122026 and 32270831 to F.R.) and Guangdong Provincial Natural Science Foundation (2023A1515010545 to H.G.).

## Author contributions

L.Z. designed the project; Z.Z., Y.W., S. J., H.C., Y.S., performed the experiments and analyzed the results; J.F., X.Y., Y.S., H.G., K.R., X.H., X.Z. provided the clinical ideas and analyses, suggestions, and samples; L.Z., F.R. wrote the manuscript.

## Competing interests

The authors declare no competing interests.

## Additional information

[1]Department of Thyroid and Neck Oncology, Key Laboratory of Cancer Prevention and Therapy, Tianjin's Clinical Research Center for Cancer, National Clinical Research Center for Cancer, The Province and Ministry Co-sponsored Collaborative Innovation Center for Medical Epigenetics, Key Laboratory of Immune

Microenvironment and Disease (Ministry of Education), Department of Biochemistry and Molecular Biology, School of Basic Medical Sciences, Tianjin Medical University Cancer Institute and Hospital, Tianjin Medical University, Tianjin, China. [2]Department of Endocrinology, Guangdong Provincial People's Hospital (Guangdong Academy of Medical Sciences), Southern Medical University, Guangzhou, Guangdong, China. [3]Department of Pathology, Tianjin Central Hospital of Gynecology and Obstetrics, Tianjin, China. [4]School of Life Sciences, Southern University of Science and Technology, Shenzhen, Guangdong, China. [5]Department of General Surgery, Tianjin Medical University General Hospital, Tianjin Medical University, Tianjin, China. [6]Department of Radiation Oncology, Key Laboratory of Cancer Prevention and Therapy, Tianjin's Clinical Research Center for Cancer, National Clinical Research Center for Cancer, Tianjin Medical University Cancer Institute and Hospital, Tianjin, China. [7]Department of Thyroid and Neck Tumor, Tianjin Medical University Cancer Institute and Hospital, National Clinical Research Center for Cancer, Key Laboratory of Cancer Prevention and Therapy, Tianjin's Clinical Research Center for Cancer, Tianjin, China. [8]Department of Thyroid and Neck Tumor, Key Laboratory of Cancer Prevention and Therapy, Tianjin's Clinical Research Center for Cancer, National Clinical Research Center for Cancer, The Province and Ministry Co-sponsored Collaborative Innovation Center for Medical Epigenetics, Key Laboratory of Immune Microenvironment and Disease (Ministry of Education), Department of Biochemistry and Molecular Biology, School of Basic Medical Sciences, Tianjin Medical University Cancer Institute and Hospital, Tianjin Medical University, Tianjin, China. [9]These authors contributed equally: Zhenlei Zhang, Yufan Wu, Jinrong Fu. ✉e-mail: guanhaixia@gdph.org.cn; raof@sustech.edu.cn; shzhaoli@tmu.edu.cn

