## [Peer Review File · Nature Communications]

Proteostatic reactivation of the developmental transcription factor TBX3 drives BRAF/MAPK-mediated tumorigenesisReviewers' comments:

Reviewer #1 (Remarks to the Author):

The manuscript titled "Bidirectional regulation of TBX3 by E3 ligase CRL5SPSB3 and deubiquitinase USP15 mediates BRAF/MAPK-directed tumorigenesis and development" by Wu et al. provides novel data to show that levels of the T-box transcription factor TBX3 is post-translationally regulated by the E3 ligase CRL5SPSB3 and the deubiquitinase USP15. Using extensive in vitro and in vivo models the authors show that USP15 directly interacts with and deubiquitinates TBX3, thereby preventing TBX3 from proteasomal degradation and stabilizing it. In addition, the authors found that this interaction occurred between the UBL1 and activation domains of USP15 and TBX3 respectively. The authors further show that the Cullin-Ring E3 ligase, CUL5, ubiquitinates TBX3 through the substrate receptor, SPSB3. While they show that the SOCS and repression 1 domains are required for SPSB3 and TBX3 interaction, they provide evidence that ubiquitylation occurs on the activation domain of TBX3 at position K466. The authors provide evidence of the relevance of the regulation of TBX3 by USP15 and CRL5SPSB3 during embryonic lineage differentiation and epithelial transformation and tumour progression. In summary, the authors present important data showing the homeostatic control of TBX3 stability by USP15 and CRL5SPSB3 which is required during development and cancer progression downstream of the BRAF/MAPK pathway and it should be accepted for publication in Nature Communications provided that the following comments are addressed:

Major comments:

1. Throughout the manuscript the authors use different cell types/cell lines which is not only confusing but it raises some concerns about conclusions that they arrive at. At the very least, the authors should provide an explanation for why they believe that doing this does not detract from their key findings.
2. Densitometric analysis should be provided for all western blots. This is especially important because there are occasions where the description of the results is not obvious in the figures.
3. The authors need to show quantification of the results from immunocytochemistry and immunohistochemistry shown in figures 1d, 3e, 3j, 5c, 6c -e.
4. Figure 1f: The data would be more convincing if the authors separated the samples on one gel and showed one exposure of the western blots.
5. Figure 1m: $\Delta D3$ also appears to reduce USP15 binding to TBX3. This should be described, and the authors should comment on why this was not further explored.
6. Figure 2g: the authors should provide a reason(s) for looking at the specific binding partners. Furthermore, the band in the IgG lane for SPBS3 should be explained.
7. Figure 2i: there is also a decrease in TBX3 when SPRY is deleted and therefore the conclusion that only the SOCS domain is important is not accurate.
8. Figure 2k: the authors infer that the TBX3 activation domain is important for binding to SPSB3, but the bands obtained for $\Delta R1$ and $\Delta R1 + \Delta A$ do not seem very different in intensity.
9. Line 252: the authors should show the recovered molecular markers in Figures 3a and 3b, and in the Extended data.
10. Figure 4: Since R values less than 0.3 or more than -0.3 are considered biologically negligible, the correlation analyses shown in b, d, j, and k are very weak. Furthermore, the correlation for the BRAFV600E mutated cancer (SKCM) is less than that for the wild type BRAF, which does not support the conclusion made by the authors.

11. Figure 4i: The authors should perform a rescue experiment (SPSB3+TBX3) as they have done in figures 3c and 4h.
12. Figures 4j and k: The authors show convincingly that SPBS3 and USP15 regulate TBX3 protein levels through ubiquitylation and deubiquitylation respectively and it is not appropriate to compare the mRNA levels of SPBS3, USP15 and TBX3 to support this as they mention in lines 305-309. The results shown in these figures address a different question and should either be described elsewhere in the manuscript or removed.
13. Fig. 5b and lines 325-327: The interchangeable use of whole thyroid tissues and tumors in the figure, figure legend and results are confusing, and this should be clarified. Furthermore, the description for each mouse group needs to be provided.
14. Figure 5d-h: As indicated in an earlier point, the authors show that USP15 regulates TBX3 levels through deubiquitylation i.e. a post-translational modification. The mechanism(s) involved in the changes in TBX3 mRNA levels observed should therefore be discussed and the manuscripts by Mowla et al., 2011 and Boyd et al., 2013 will be worth checking.
15. Figure 5e: Contrary to the conclusion drawn by the authors, lane 5 shows that TBX3 levels increase after treatment with 10 μ M ERK1/2i. This is an example of where densitometry is essential.
16. Line 355-357: the authors conclude that BRAFV600E promotes tumor development through elevating USP15 levels and consequently increased levels of TBX3. The possibility that TBX3 levels also increase due to transcriptional upregulation downstream of BRAFV600E (see Boyd et al., 2013) should be discussed.
17. Figure 6c: How do the authors explain the band for USP15 in the USP15^{-/-} thyroid.
18. The authors should emphasise the translational potential of their findings in the abstract and discussion. Furthermore, are there commercially available USP15 inhibitors that could be used in combination with current thyroid carcinoma treatments?
19. In the discussion, the authors should comment on additional pathways that have been described to regulate TBX3 post-translationally. For example, Peres et al., (2015), Willmer et al., (2015) and Sims et al., (2020).

Minor comments:

- The strain of mice used throughout the study should be stated.
- Figure 1i: The authors should label the UBL regions UBL1 and UBL2.
- Line 233: “to identify specific sites in SPSB3”, the authors should change SPSB3 to TBX3.
- Line 345-347: the sentence “MAPK directs USP15 expression through transcriptional regulation, since repression of downstream effector AP-1 protein also reduced TBX3 at a protein and mRNA levels” is confusing and needs to be rewritten.
- Line 358-359: The title is confusing and should be rewritten.
- Line 427: A reference is needed.
- Line 427-431: The sentence is confusing and needs to be rewritten.
- Figure 5b: the y axis should read “thyroid weight” and not “tumour weight”.
- The number of times that all experiments were performed should be indicated.

Reviewer #2 (Remarks to the Author):

The manuscript by Wu et al builds upon previous work from the same group (Zhang et al, Nature Communications, 2022) and explores the role of developmental transcription factor TBX3 in BRAFV600E-induced thyroid tumorigenesis. In this work, the authors identified two nodes with opposing effects on TBX3: CRL5-SPSB3, an E3 ligase that promotes TBX3 degradation, and USP5, a deubiquitinase which inhibits this process. The manuscript

includes a series of well-designed experiments using a variety of in vitro and in vivo models. The results generally support their hypotheses. The amount of experimental data is massive and at times overwhelming, suggesting that prioritization of experimental procedures and/or data display would be desirable, as the manuscript is sometimes hard to follow and risks feeling unfocused. This reviewer's main concerns are listed below:

- I miss a more clear connection with the author's previous manuscript (Zhang et al 2022). In that work, particularly in the in vivo portion, the authors knocked down Tbx3 and observed infiltration by MDSC and a role for NF-kB signaling. Here, primarily via Usp5 knockouts, which in turn lead to Tbx3 suppression, they focus on an entirely different set of molecular consequences. Since in both works they are using the same BrafV600E background as models of thyroid tumorigenesis, having some kind of overarching hypothesis that reconciles both mechanistic underpinnings would be helpful.
- In this reviewer's opinion, the experiments assessing the roles of Tbx3 and Usp15 in embryogenesis/thyroid morphogenesis are beyond the scope of this manuscript, as they do not necessarily connect well with what it is supposed to be the main message of the paper (i.e., their roles in BRAF-induced thyroid tumorigenesis). This manuscript is already data-packed, with every figure having lots of panels, plus extensive supplemental information. I would recommend deleting, reducing and refocusing. In addition, I am not sure whether physiological activation of MAPK pathway in embryogenesis vs. pathological (i.e., ligand-unresponsive and continuous) activation in tumorigenesis are comparable processes.
- Other comments:
 - It is not always clear whether HEK293T (probably chosen because they are easier to transfect) or thyroid cancer cells (preferred models) were used in in vitro experiments (e.g., Fig 2). Please clarify, and if possible, prioritize observations that were identified in BRAFV600E mutant thyroid cancer cells, as their biology is disrupted by MAPK constitutive activation and different than other lineages.
 - The choice of A375 melanoma cells for the experiments displayed in Figure 4 is highly questionable.
 - Figure 7 is not very informative in its current form. I would remove the illustrations on embryogenesis, and possibly those on tumorigenesis, and focus instead on a simplified version in which the opposing roles of CRL5 and USP15 on TBX3 regulation are displayed, based on the experimental data from this manuscript.
 - The Introduction feels at times as systematic review instead of an introduction to the essential topics that are necessary to understand and justify the experimental approach adopted here.
 - There might be multiple roles of MAPK activation in development, but their main role in thyroid cancer pathogenesis is unequivocal: promoting cell proliferation. The authors touch on this topic here and there, but it would be helpful to include experiments that delineate the effects of MAPK which are dependent vs. independent of TBX3 biology.
 - There are several grammatical errors throughout the manuscript, so I would recommend the authors to perform a thorough revision of the text.

Reviewer #3 (Remarks to the Author):

This paper claims to have identified a substrate of the deubiquitinase USP15 that may have relevance to cancer therapy. There is no conceptual advance, but the findings will be of interest to specialist readers and to the Pharma drug discovery world. It is an exhausting read with over 100 figure panels (I counted 66 in the main text). I think this is both unfair on the reviewer but also the prospective reader. It is incumbent on the authors to design well

controlled and better quantitated experiments that distill down their essential message. I think they fail to do that here.

Major comments:

All experiments involving over-expression of USP15 are flawed in that they miss an essential control. To show specificity another USP family member DUB should be expressed at similar levels. Most obvious here would be the closest family members USP4 or USP11. Otherwise any effects of USP15 over-expression could be interpreted as simple mass action effects.

The starting point is the co-IP of USP15 following over-expression of TBX3 in HEX293T cells. The over-expression must be enormous to visualise TBX3 and USP15 directly with what looks like a silver stained gel. In effect this experiment has been performed previously as part of a systematic analysis of protein-protein interactions, Bioplex, by the Harper/Gygi labs. In their case using either partner as bait- the other one was not identified. This has to make me somewhat skeptical of this foundational experiment.

Some of my other comments

Fig. 1d- does not show colocalisation- simply that both are in the nucleus.

Fig 1f- loading needs to be adjusted so that starting values, time 0, are more similar.

Fig 1h -representative of several others. The MW of TBX3 is close to 95kD- yet the effect of USP15 overexpression is to reduce an entire ubiquitin smear, including the MW range below 95kD.

Fig 6B quantitation does not seem to match the data presented. More sample need to loaded for -/- condition. How can you get an apparent increase in TBX3 after 2 hours cyclohex.

Discussion of pre-existing USP15 literature is weak.

Re: Manuscript NCOMMS-23-08495-T

Thank you for your letter regarding the above submitted manuscript titled “Bidirectional regulation of TBX3 by E3 ligase CRL5^{SPSB3} and deubiquitinase USP15 mediates BRAF/MAPK-directed tumorigenesis and development”, which was revised for publication on Nature Communications.

We are very grateful for the helpful comments and criticisms from the reviewers. We have undertaken revisions in response to the reviewers’ comments and would now like to submit a revised manuscript. The detailed responses and revisions are given below:

Reviewer #1 (Remarks to the Author):

The manuscript titled “Bidirectional regulation of TBX3 by E3 ligase CRL5^{SPSB3} and deubiquitinase USP15 mediates BRAF/MAPK-directed tumorigenesis and development” by Wu et al. provides novel data to show that levels of the T-box transcription factor TBX3 is post-translationally regulated by the E3 ligase CRL5^{SPSB3} and the deubiquitinase USP15. Using extensive in vitro and in vivo models the authors show that USP15 directly interacts with and deubiquitinates TBX3, thereby preventing TBX3 from proteasomal degradation and stabilizing it. In addition, the authors found that this interaction occurred between the UBL1 and activation domains of USP15 and TBX3 respectively. The authors further show that the Cullin-Ring E3 ligase, CUL5, ubiquitinates TBX3 through the substrate receptor, SPSB3. While they show that the SOCS and repression 1 domains are required for SPSB3 and TBX3 interaction, they provide evidence that ubiquitylation occurs on the activation domain of TBX3 at position K466. The authors provide evidence of the relevance of the regulation of TBX3 by USP15 and CRL5^{SPSB3} during embryonic lineage differentiation and epithelial transformation and tumour progression. In

summary, the authors present important data showing the homeostatic control of TBX3 stability by USP15 and CRL5^{SPSB3} which is required during development and cancer progression downstream of the BRAF/MAPK pathway and it should be accepted for publication in Nature Communications provided that the following comments are addressed:

Response: Thank the reviewer for the encouraging comments and very helpful suggestions. Based on our continuous study for TBX3, our group is quite curious about whether similar functions and regulations work through TBX3-directed embryonic and tumorigenic processes. According to these thoughts and other reviewers' comments, also to make the study more focused, we removed the SPSB3 part and explored more on USP15-mediated TBX3 stabilization.

Mainly, we systematically confirmed the biological significance of USP15-TBX3 axis during BRAF/MAPK activity-determined development and tumorigenesis. Through comparing the histological characteristics as well as molecular profilings between Tbx3 or Usp15 knockout developmental and tumorigenic models, we actually found that loss of either gene leads to comparable tumor redifferentiation, which parallels the overdifferentiation tendency during development. The clinical relevance is highlighted in that USP15 and TBX3 are highly correlated with BRAFV600E signature and poor tumor prognosis within different types of BRAFV600E cancers, at both transcriptomic and proteomic levels. With the additional molecular, genetic, and clinical verifications, we realize that the stability control of TBX3 by USP15 represents a critical proteostatic mechanism downstream of BRAF/MAPK-directed developmental homeostasis and pathological transformation. Our findings support that tumorigenesis largely relies on epithelial dedifferentiation achieved via embryonic regulatory program reinitiation. In this situation, we title the manuscript as "Proteostatic reactivation of the developmental transcription factor TBX3 drives BRAF/MAPK-mediated tumorigenesis".

As I mentioned and documented in the initial version, the CRL5^{SPSB3}-mediated TBX3 degradation is surely an important and novel finding in the field. We have also gained more progresses regarding the biological significance of this mechanism as well as other involved modification events. We look forward to present the story in the near future. Hope you will understand the reorganization and provide more suggestions.

Major comments:

1. Throughout the manuscript the authors use different cell types/cell lines which is not only confusing but it raises some concerns about conclusions that they arrive at. At the very least, the authors should provide an explanation for why they believe that doing this does not detract from their key findings.

Response: Thank the reviewer for the kind question. Yes, we used a series of tumor cell lines in the beginning to illustrate that ubiquitin-proteasome system (UPS)-dependent degradation of TBX3 occurs commonly, although with different rates (supplementary Fig 1a).

In the following biochemical experiments, we performed some exogenous interaction

and deubiquitylation assays using HEK293T as a well-accepted tool cell line. To testify the physiological interaction and function under BRAFV600E-induced tumorigenesis, we selected BRAFV600E-mutated thyroid cancer cells as well as melanoma cells. Moreover, to check the significance of USP15-TBX3 axis during BRAFV600E-induced successive transformation program, we build the *in vitro* transformation model by introducing BRAFV600E into the normal thyrocyte line Nthy-ori 3-1. In the revised version, we clear the contexts for different cell line application to avoid unnecessary distraction.

2. Densitometric analysis should be provided for all western blots. This is especially important because there are occasions where the description of the results is not obvious in the figures.

Response: Thank you for the suggestion. We have provided densitometric analyses to all the western blot results. Hope the quantification will help us appreciate the results better.

3. The authors need to show quantification of the results from immunocytochemistry and immunohistochemistry shown in figures 1d, 3e, 3j, 5c, 6c -e.

Response: Thank you for the suggestion. We have quantified the related immunocytochemistry and immunohistochemistry results as well as supplementary results. Please allow us to detail the re-organized figures as followed: current Supplementary Fig. 1c corresponds to previous Fig 1d; current Fig 3e corresponds to previous Fig 3e; current Fig 3j corresponds to previous Fig 5c; current Fig 6c-7d corresponds to previous Fig 6c-6e. Hope the quantification will help us appreciate the results better.

4. Figure 1f: The data would be more convincing if the authors separated the samples on one gel and showed one exposure of the western blots.

Response: The previous Fig 1f result is now presented as Fig 1i. As the reviewer suggest, we have run the samples on the same gel and showed one exposure of the western blot, which further confirmed that USP15 knockdown shortens the half-life of TBX3. And also, we are pretty sure that all the western blot bands of Cycloheximide (CHX) pulse-chase assays come from the same exposure intensity between control and experimental groups.

5. Figure 1m: $\Delta D3$ also appears to reduce USP15 binding to TBX3. This should be described, and the authors should comment on why this was not further explored.

Response: The previous Fig 1m result is now presented as Fig 2f. We had the same impression as the reviewer at the beginning. However, $\Delta D3$ does not always show reduced binding with TBX3 through different replicates, distinct from $\Delta U1$. During the revision, we double-checked the effect of $\Delta U1$ on TBX3 ubiquitylation via *in vivo* deubiquitylation assays and found that $\Delta U1$ could no longer deubiquitylate TBX3 (Fig 2h). Together, these data all point to the necessity of U1 in mediating the USP15 binding with TBX3.

6. Figure 2g: the authors should provide a reason(s) for looking at the specific binding partners. Furthermore, the band in the IgG lane for SPBS3 should be explained.

Response: As mentioned in the beginning, we removed the SPSB3 part in the revised manuscript, but we are happy to discuss with the reviewer regarding this result. We check the components for CRL5 complex in the immuno-precipitates with TBX3 to confirm if SPSB3-mediated binding with TBX3 involves CRL5 complex, which is the common mechanism how most receptors interact with and lead the substrates to degradation. Indeed, critical components are pulled-down by TBX3, indicating that the degradation truly depends on CRL5^{SPSB3} complex¹.

Furthermore, we performed more repeats and obtained clearer background of Co-IPs with no bands in the IgG lanes.

7. Figure 2i: there is also a decrease in TBX3 when SPRY is deleted and therefore the conclusion that only the SOCS domain is important is not accurate.

Response: Thank you for kindly pointing this out. The Δ SPRY truncation does not always reduce the binding with TBX3 as significantly as Δ SOCS. Referring to the working mechanism employed by SPSB3 with other substrates, we conclude that SOCS domain is the predominant domain that mediates binding with TBX3. However, we may not exclude the possible involvement of SPRY in this case, and we will check further and make more accurate conclusion.

8. Figure 2k: the authors infer that the TBX3 activation domain is important for binding to SPSB3, but the bands obtained for Δ R1 and Δ R1 + Δ A do not seem very different in intensity.

Response: We conducted the experiments a couple of times and spent a large effort to propose the molecular binding model. First, deletion alone of activation domain makes no much impact on interaction, but the ubiquitylation and degradation is severely affected, indicating that the activation domain is a vital region for ubiquitylation. On the other side, deletion alone of R1 domain abrogates the binding significantly, the ubiquitylation and degradation happens even more, indicating that R1 domain is more required for binding. As the reviewer suggested, we quantified the bands by densitometric analysis and realized that further deletion of activation domain [Δ (A+R1)] affects the interaction more than Δ R1 alone. Taken together, we speculate that the conformation formed by activation and R1 domains may be necessary for TBX3 to be recognized and bound by SPSB3, while the R1 domain may block activation domain to be ubiquitylated by SPSB3.

9. Line 252: the authors should show the recovered molecular markers in Figures 3a and 3b, and in the Extended data.

Response: As mentioned in the beginning, we removed the SPSB3 part in the revised manuscript, but we will definitely show the recovered molecular markers in Figures 3a and 3b and Extended data in the future SPSB3 paper. Appreciate the reviewer for the suggestion.

10. Figure 4: Since R values less than 0.3 or more than -0.3 are considered biologically negligible, the correlation analyses shown in b, d, j, and k are very weak. Furthermore, the correlation for the BRAFV600E mutated cancer (SKCM) is less than that for the wild type BRAF, which does not support the conclusion made by the authors.

Response: Thank the review for pointing this out. In the current manuscript, we focus more on the biological significance of USP15-TBX3 axis during BRAF/MAPK activity-determined development and tumorigenesis. For this purpose, we explore more about the direct regulation between BRAF/MAPK activity and USP15 abundance, the important role of USP15 in mediating BRAFV600E-induced transformation, as well as the clinical relevance of our findings.

Firstly, we actually find activated BRAF/MAPK pathway directly upregulates USP15 expression through AP-1 factors-mediated transcriptional activation, which occurs through different types of BRAFV600E-mutated cancer cells (Figure 4). Secondly, we confirm that the dosage of USP15-TBX3 axis is critical for BRAF/MAPK activity-determined differentiation states (Figure 7). Thirdly, we characterize the clinical significance of USP15-TBX3 regulation by analyzing tissue-arrays and public data for thyroid cancer and melanoma in which BRAFV600E is the most dramatic carcinogenic mutation (Figure 5).

In details, within thyroid cancer tissue-arrays, the positive correlation between USP15 and TBX3 may not be significant enough (current Fig 5a & 5b, corresponding to previous supplementary Fig 5b, $R=0.6668$; current Fig 5c & 5d, corresponding to previous Fig 4c and 4d, $R=0.4403$). Whereas, USP15 expression in BRAFV600E-positive samples is significantly higher than that in BRAFWT samples (Fig 5f and 5g). The melanoma tissue-array stainings show even higher positive correlation between the two factors (Fig 5k and 5l). More supportively, the thyroid carcinoma proteomics data from iProX also confirm that USP15 protein level positively correlated with that of TBX3 among samples with both factors detectable (Fig 5m). As the TCGA data are concerned, USP15 positively correlates with that of both TBX3 and BRAF (current supplementary Fig 4g, corresponding to previous Fig 4j and 4k). Remarkably, once we sort out BRAFV600E mutated patients, the positive correlation between USP15 with TBX3 or with BRAF in both THCA and SKCM was significantly increased (supplementary Fig 4h).

All these new data and analyses provide reliable evidences for the function of USP15-TBX3 under BRAF/MAPK activity.

Revisions have been made on Figures 4,5,7, supplementary Figure 4, as well as on page 10, line 255 to line 260: Another cohort of tissue array composed of PTC and the adjacent tissues with accessible BRAF genetic information ($n=93$) was analyzed by serial sectioning. Overall, USP15 level still positively correlates with TBX3 level (Fig. 5c, d). Quite interestingly, USP15 expression in BRAF^{V600E}-positive samples is significantly higher than that in BRAF^{WT} group (Fig. 5e, f), causing 82.1% of USP15^{high} expression in BRAF^{V600E} samples in comparison to 48.6% in BRAF^{WT} samples (Fig. 5g).

page 11, line 279 to line 293: Our findings were further supported by available thyroid carcinoma proteomics data from iProX, where USP15 protein level positively correlated with that of TBX3 among samples with both factors detectable (Fig. 5m, n=52). Based on THCA transcriptome data from The Cancer Genome Atlas (TCGA), USP15 expression is positively correlated with TBX3, even BRAF, consistent with the high correlation between TBX3 and BRAF (Supplementary Fig. 4g). Similarly, USP15 positively correlates with BRAF in melanoma (named SKCM) where TBX3 has been confirmed to be highly pro-tumorigenic. USP15 and TBX3 transcriptional expressions also show positively correlated tendency, although not statistically significant (Supplementary Fig. 4g, h). Remarkably, once we sort out BRAF^{V600E} mutated patients, the positive correlation between USP15 with TBX3 or with BRAF in both THCA and SKCM was significantly increased (Supplementary Fig. 4h). These analyses indicate that high USP15 expression level could predict higher tumor progression and poorer overall survival, in BRAF^{V600E} related malignancies, presumably due to its regulatory function on TBX3 protein homeostasis.

11. Figure 4i: The authors should perform a rescue experiment (SPSB3+TBX3) as they have done in figures 3c and 4h.

Response: Thank the reviewer for the suggestion. We have performed a rescue experiment by overexpressing SPSB3+TBX3, which efficiently recovered the tumor growth. We will include this part of result in the future SPSB3 paper.

12. Figures 4j and k: The authors show convincingly that SPBS3 and USP15 regulate TBX3 protein levels through ubiquitylation and deubiquitylation respectively and it is not appropriate to compare the mRNA levels of SPBS3, USP15 and TBX3 to support this as they mention in lines 305-309. The results shown in these figures address a different question and should either be described elsewhere in the manuscript or removed.

Response: Based on our findings that activated BRAF/MAPK pathway directly upregulates both USP15 and TBX3 expression through AP-1 factors-mediated transcriptional activation, it's supportive to detect positive correlation between USP15 and TBX3, as well as BRAF in TCGA data² (current supplementary Fig 4g and 4h, corresponding to previous Fig 4j and 4k). Further, the thyroid carcinoma proteomics data from iProX also show that USP15 protein level positively correlates with that of TBX3, which highly supports our mechanism finding that USP15 stabilizes TBX3 protein (Fig 5m).

Revisions have been made on Figure 5, supplementary Figure 4, as well as on page 11, line 279 to line 293: Our findings were further supported by available thyroid carcinoma proteomics data from iProX, where USP15 protein level positively correlated with that of TBX3 among samples with both factors detectable (Fig. 5m, n=52). Based on THCA transcriptome data from The Cancer Genome Atlas (TCGA), USP15 expression is positively correlated with TBX3, even BRAF, consistent with the high correlation between TBX3 and BRAF (Supplementary Fig. 4g). Similarly, USP15 positively correlates with BRAF in melanoma (named SKCM) where TBX3 has been confirmed to be highly pro-tumorigenic. USP15 and TBX3 transcriptional expressions also show positively correlated tendency, although not statistically

significant (Supplementary Fig. 4g, h). Remarkably, once we sort out BRAF^{V600E} mutated patients, the positive correlation between USP15 with TBX3 or with BRAF in both THCA and SKCM was significantly increased (Supplementary Fig. 4h). These analyses indicate that high USP15 expression level could predict higher tumor progression and poorer overall survival, in BRAF^{V600E} related malignancies, presumably due to its regulatory function on TBX3 protein homeostasis.

13. Fig. 5b and lines 325-327: The interchangeable use of whole thyroid tissues and tumors in the figure, figure legend and results are confusing, and this should be clarified. Furthermore, the description for each mouse group needs to be provided.

Response: We have clarified the descriptions as the reviewer mentioned, especially in the current Fig 3h, corresponding to previous Fig 5b.

14. Figure 5d-h: As indicated in an earlier point, the authors show that USP15 regulates TBX3 levels through deubiquitylation i.e. a post-translational modification. The mechanism(s) involved in the changes in TBX3 mRNA levels observed should therefore be discussed and the manuscripts by Mowla et al., 2011 and Boyd et al., 2013 will be worth checking.

Response: We agree with the reviewer that TBX3 is subjected to different regulations during tumorigenesis. Besides what we present here and before², TBX3 was up-regulated by PMA in an AP-1 dependent manner. Moreover, TBX3 was shown to work downstream of BRAFV600E-induced melanoma transformation to repress E-cadherin expression. We incorporated these mechanism findings into our introduction and discussion.

Revisions have been made on introduction and discussion on page 3, line 66 to line 68: Recently, the oncogenic function of TBX3 draws attention, where TBX3 is often up-regulated in tumorigenesis and promotes tumor cell proliferation and metastasis, especially during BRAF/MAPK induced tumorigenesis.

page 16, line 421 to line 423: Multiple oncogenic pathways have been found to regulate TBX3 expression in occurrence of melanoma, lung, thyroid, breast and other types of cancers, here we define USP15 as a critical stabilizer for TBX3.

15. Figure 5e: Contrary to the conclusion drawn by the authors, lane 5 shows that TBX3 levels increase after treatment with 10 μ M ERK1/2i. This is an example of where densitometry is essential.

Response: We apologize for not providing the most representative results. We have replaced the panel and added densitometric analysis result. The previous Fig 5e result is now presented as Fig 4b.

16. Line 355-357: the authors conclude that BRAFV600E promotes tumor development through elevating USP15 levels and consequently increased levels of TBX3. The possibility that TBX3 levels also increase due to transcriptional upregulation downstream of BRAFV600E (see Boyd et al., 2013) should be discussed.

Response: We agree with the reviewer and never ignore the transcriptional

upregulation of TBX3 downstream of BRAFV600E. As reported by Boyd et al, TBX3 functions downstream of BRAFV600E and represses the expression of E-cadherin³, we reveal quite similar regulation within BRAFV600E-induced thyroid carcinogenesis². Further, the USP15 stabilizes the TBX3 protein and functions also downstream of BRAF/MAPK, since loss of USP15 blocked BRAFV600E-induced TBX3 acceleration (Fig 7f). In contrast, repression of TBX3 by BRAF/ERK inhibitors is partially reversed by MG132, indicating that USP15 loss caused TBX3 degradation acceleration accounts part of the overall TBX3 reduction (Fig 4c and supplementary Fig 3b). Therefore, we prefer the rationale that activated BRAF/MAPK pathway achieves high TBX3 expression through multi-levels of regulations. In addition to introduction regarding the transcriptional upregulation downstream of BRAFV600E for TBX3, we also revise the description in the result conclusion.

Related revisions have been made on Figure 4 and Figure 7, as well as on page

14, line 361 to line 372: The next question is whether Usp15-stabilized Tbx3 expression is the determinant for BRAF^{V600E}-induced de-differentiation. We first took advantage of the Nthy/BRAF^{V600E} model to monitor the successively pathophysiological transition. Indeed, BRAF^{V600E} effectively initiates the follicular de-differentiation, represented with dramatically reduced thyroid lineage factors. However, the de-differentiation was reversed by USP15 knockdown and accompanying TBX3 reduction (Fig. 7f and Supplementary Fig. 5e). At the early BRAF^{V600E}-induced transformation and de-differentiation *in vivo*, E16.5 mPTC, the initial proliferation could no longer pursue once Usp15 is absent, with so caused Tbx3 loss (Fig. 7g, Supplementary Fig. 5f, g).

page 9, line 219 to line 224: MG132 treatment rescued TBX3 protein loss under BRAF or ERK1/2 inhibitor, indicating that USP15 loss caused TBX3 degradation acceleration accounts part of the overall TBX3 reduction (Fig. 4c and Supplementary Fig. 3b). Since repression of downstream effector AP-1 protein directly reduced USP15 mRNA as well as protein levels (Fig. 4a, d and Supplementary Fig. 3b), we hypothesized that MAPK directs USP15 expression through transcriptional regulation.

17. Figure 6c: How do the authors explain the band for USP15 in the USP15/- thyroid.

Response: We check the knock-out efficiency through different litters and present a more representative result in Fig 6d and replace previous Fig 6c.

18. The authors should emphasise the translational potential of their findings in the abstract and discussion. Furthermore, are there commercially available USP15 inhibitors that could be used in combination with current thyroid carcinoma treatments?

Response: Thank the reviewer for the suggestion. Since our revised manuscript focuses more on the biological significance of USP15-TBX3 axis during BRAF/MAPK activity-determined development and tumorigenesis. We address the translational potential of our findings in the abstract and discussion.

Regarding the USP15 inhibitors, there were no commercially available inhibitors during our initial submission. Luckily, we get access to a research group who has

patented some potential inhibitors against USP15 (Chinese invention patent application: CN201810515378.3). Due to the limitation of time and animal resources, we check their effect on BRAFV600E-mutated tumor cells. As showed below, a couple of inhibitors elevated the sensitivity of tumor cells to BRAF inhibitor Vemurafenib treatment. We will definitely push forward the translation of our findings.

[Redacted]

Figure 1. Dose-response curve of BRAF inhibitor Vemurafenib, USP15 inhibitor and combination medications in thyroid carcinoma cell lines K1, 8505C and melanoma cell line A375.

Table 1. IC50 of BRAF inhibitor Vemurafenib, USP15 inhibitor and combination medications in thyroid carcinoma cell lines K1, 8505C and melanoma cell line A375.

[Redacted]

Related revisions have been made on discussion on page 17, line 441 to line 453: We and other groups have converged on the critical oncogenic function of TBX3 in various kinds of tumors. While the transcription factor feature makes TBX3 hard to be targeted, the current findings open up the possibility to modulate TBX3 expression through intervening the enzyme activity of USP15 or the physiological molecular binding. Meanwhile, with the progress of PROTAC technique and definition of specific degradation machinery, it will be possible to accurately modulate the endogenous level of TBX3. Therefore, this study uncovers the dedifferentiation mechanisms under BRAF/MAPK-induced tumorigenesis based on defining the function of USP15 in homeostatic TBX3 regulation during development and tumorigenesis. Translationally, the findings shed light on targeting strategies for USP15-TBX3 axis related developmental and oncogenic diseases and provide new thoughts for BRAF/MAPK related therapy designs.

19. In the discussion, the authors should comment on additional pathways that have been described to regulate TBX3 post-translationally. For example, Peres et al., (2015), Willmer et al., (2015) and Sims et al., (2020).

Response: Thank the reviewer for the suggestion. We have re-organized our discussion and commented on additional pathways that have been described to regulate TBX3 post-translationally. The findings of Peres et al., (2015), Willmer et al., (2015) and Sims et al., (2020)., Yano et al (2011) have been included. Clearly, there is still a large scientific void about the post-translational regulations of TBX3, it will be

necessary to explore this field.

Related revisions have been made on discussion on page 16, line 421 to line

432: Multiple oncogenic pathways have been found to regulate TBX3 expression in occurrence of melanoma, lung, thyroid, breast and other types of cancers, here we define USP15 as a critical stabilizer for TBX3. Due to the conflicting functions of substrates, USP15 shows dual effects in tumorigenesis. For instance, USP15 copy number is increased in glioblastoma, breast cancer, and ovarian cancer, but decreased in pancreatic cancer. The current finding provides more evidence about the USP15 oncogenic function, especially during BRAF^{V600E} mutated cancers. Moreover, a few kinases participate in phosphorylation of TBX3, such as AKT3 at S720, cyclin A-CDK2 at either S190 or S354, p38 MAP kinase at S692, and AKT1 is also involved. Some of the phosphorylation sites also have effect on the stability of TBX3, it will be interesting to check the functional correlation between different types of modifications.

Minor comments:

- The strain of mice used throughout the study should be stated.

Response: Sure. We have stated the mouse strains clearly in our figures and text, mainly Fig 3g and supplementary Fig 2h.

Related revisions have been made on figure legends on page 524, line 525: The transgenic mice crossing strategy of mPTC and mPTC/Us^{p15}^{-/-} (TPO-Cre/Braf^{V600E}/Us^{p15}^{-/-}) mice.

- Figure 1l: The authors should label the UBL regions UBL1 and UBL2.

Response: Sure. We have corrected the label in current Fig 2e, corresponding to previous Fig 1l.

- Line 233: “to identify specific sites in SPSB3”, the authors should change SPSB3 to TBX3.

Response: Thank you for pointing this out. We will definitely correct the writing and be more careful in the potential SPSB3 story.

- Line 345-347: the sentence “MAPK directs USP15 expression through transcriptional regulation, since repression of downstream effector AP-1 protein also reduced TBX3 at a protein and mRNA levels” is confusing and needs to be rewritten.

Response: Thank you for pointing this out. We have re-organized this part and rewritten the description.

Related revisions have been made on figure legends on page 9, line 216 to line

224: Comparably, pharmacological inhibition of BRAF with PLX4032, or inhibition of ERK1/2 with SCH772984 significantly down-regulated USP15 expressions in BRAF^{V600E}-mutated human thyroid cancer cells (Fig. 4b and Supplementary Fig. 3a). MG132 treatment rescued TBX3 protein loss under BRAF or ERK1/2 inhibitor, indicating that USP15 loss caused TBX3 degradation acceleration accounts part of the overall TBX3 reduction (Fig. 4c and Supplementary Fig. 3b). Since repression of downstream effector AP-1 protein directly reduced USP15 mRNA as well as protein levels (Fig. 4a, d and Supplementary Fig. 3b), we hypothesized that MAPK directs USP15 expression through transcriptional regulation.

- Line 358-359: The title is confusing and should be rewritten.

Response: Thank you for pointing this out. We have rewritten the title into “USP15 coordinates TBX3 abundance through BRAF/MAPK activity-determined development and tumorigenesis”.

- Line 427: A reference is needed.

Response: Thank you for the suggestion. References 1-3 suit here very much and have been added.

- Line 427-431: The sentence is confusing and needs to be rewritten.

Response: Thank you for pointing this out. We have rewritten the discussion for this part.

Related revisions have been made on discussion on page 15, line 392 to line

400: Here we observe that the regained differentiation behaviors upon Usp15 or Tbx3 knockout in BRAF^{V600E}-induced tumorigenesis, especially the restored thyroid hormone levels, Nis, Tpo, Tg and other differentiation factor expressions, parallel with the elevated differentiation features in Usp15- or Tbx3-knockout organs. Therefore, BRAF/MAPK pathway up-regulates developmental factor TBX3 for tumorigenesis by interfering with the proteostasis module in addition to transcriptional regulation, which could be re-activation of the developmental program how BRAF/MAPK intermingles with lineage differentiation factors.

- Figure 5b: the y axis should read “thyroid weight” and not “tumour weight”.

Response: Thank you for pointing this out. We have corrected the y axis definition in the current Fig 3h, corresponding to previous Fig 5b.

- The number of times that all experiments were performed should be indicated.

Response: Yeah, We indicate the number of times that experiments were performed when necessary. We totally understand we can never make conclusions based on single or limited times of experiments, although we normally show representative results.

Reviewer #2 (Remarks to the Author):

The manuscript by Wu et al builds upon previous work from the same group (Zhang et al, Nature Communications, 2022) and explores the role of developmental transcription factor TBX3 in BRAFV600E-induced thyroid tumorigenesis. In this work, the authors identified two nodes with opposing effects on TBX3: CRL5-SPSB3, an E3 ligase that promotes TBX3 degradation, and USP5, a deubiquitinase which inhibits this process. The manuscript includes a series of well-designed experiments using a variety of in vitro and in vivo models. The results generally support their hypotheses. The amount of experimental data is massive and at times overwhelming, suggesting that prioritization of experimental procedures and/or data display would be desirable, as the manuscript is sometimes hard to follow and risks feeling unfocused. This reviewer's main concerns are listed below:

- I miss a more clear connection with the author's previous manuscript (Zhang et al 2022). In that work, particularly in the in vivo portion, the authors knocked down Tbx3 and observed infiltration by MDSC and a role for NF- κ B signaling. Here, primarily via Usp5 knockouts, which in turn lead to Tbx3 suppression, they focus on an entirely different set of molecular consequences. Since in both works they are using the same BrafV600E background as models of thyroid tumorigenesis, having some kind of overarching hypothesis that reconciles both mechanistic underpinnings would be helpful.

Response: In our first paper on *Oncogene*⁴, we found TBX3 promotes thyroid cancer cell proliferation *in vitro* and xenograft models, mostly through transcriptionally repressing p57^{KIP2} by recruiting PRC2 components. Further, we confirmed the *in vivo* function of TBX3 in the mPTC (the genetic model TPO-Cre; BRAFV600E) models, and we found TBX3 participates in establishing immune-suppressive tumor microenvironment (TME), mostly through elevating IKK β /NF- κ B signaling thus MDSCs recruitment².

Back to this manuscript, to understand whether Usp15 ko affects mPTC development via down-regulating Tbx3 expression and related functions, we also checked infiltration of MDSCs in the mutant TME in addition to p57^{KIP2}/proliferation^{2, 4}. Indeed, mPTC/Usp15 ko mutants exhibit not only reduced proliferation/increased p57^{KIP2}, but also attenuated MDSCs infiltration (Fig 3j).

In the revised version, we pushed forward the phenotype characterization between Tbx3 and Usp15 knockout tumorigenic models. Through analyzing the histological characteristics and molecular profilings in parallel, we actually found that loss of either gene leads to comparable tumor re-differentiation, further supporting their functional dependency. During both *in vitro* BRAFV600E-induced human thyrocyte Nthy-ori 3-1 transformation and *in vivo* embryonic mPTC model, we both find that USP15 is required for TBX3 elevation and tumor initiation (Fig 7). Based on our current and previous studies, BRAF/MAPK signaling up-regulates TBX3 level through both direct transcriptional activation and USP15-mediated post-translational stabilization, thus

TBX3 plays a pivotal role downstream of BRAF/MAPK-induced events and needs to be finely controlled.

- In this reviewer's opinion, the experiments assessing the roles of Tbx3 and Usp15 in embryogenesis/thyroid morphogenesis are beyond the scope of this manuscript, as they do not necessarily connect well with what it is supposed to be the main message of the paper (i.e., their roles in BRAF-induced thyroid tumorigenesis). This manuscript is already data-packed, with every figure having lots of panels, plus extensive supplemental information. I would recommend deleting, reducing and refocusing. In addition, I am not sure whether physiological activation of MAPK pathway in embryogenesis vs. pathological (i.e., ligand-unresponsive and continuous) activation in tumorigenesis are comparable processes.

Response: Thank the reviewer for such helpful suggestions and we apologize for not presenting the findings in a more focused way. As other groups and we have found, TBX3 functions as an instructive factor through multiple organ developments and tumor formations. We are eager to define the proteostasis regulators of TBX3 to push mechanism study and targeting translation, since scientists all have realized the important roles of deregulated proteostatic controls of key factors through malignancies. When we discover the indispensable role of USP15-TBX3 stabilization axis downstream of BRAFV600E-induced tumorigenesis, we are very curious of the physiological meaning of the same regulation in organ development since BRAF/MAPK pathway are usually active during early embryogenesis progenitors amplification.

Indeed, we observed highly comparable developmental defects in Usp15 ko and Tbx3 ko organogenesis. Taken the thyroid as an example, both mutants present overdifferentiation tendency and form precociously defected follicles, evaluated by histological and transcriptomic profilings (Fig 7A). As reported, p-Erks participate in the specification of early embryonic endoderm foregut⁵, which is the precursor for thyroid. Considering together with the dynamic and synchronistical expressions between p-Erks, Usp15, and Tbx3, we propose that Usp15 could coordinate Tbx3 abundance through BRAF/MAPK activity-determined development (Figure 6 and 7).

Then, from physiological to pathological states, BRAFV600E-induced transformation also relies on Usp15-maintained Tbx3 protein through different species (Fig 7f and 7g). Moreover, the clinical significance becomes more convincing when we realize the correlated protein abundances of USP15 and TBX3 in the clinical proteomics and tissue arrays, across thyroid cancer and melanoma samples (Fig 5 and supplementary Fig 4). Most encouragingly, loss of either gene leads to comparable tumor redifferentiation, which parallels aforementioned overdifferentiation tendency during development.

With the additional molecular, genetic, and clinical verifications, we conclude that activated BRAF/MAPK pathway by BRAFV600E mediates malignant transformation by reactivating Usp15-stabilized Tbx3. Our findings support that tumorigenesis largely relies on epithelial dedifferentiation achieved via embryonic regulatory program reinitiation. Based on the reviewer and editor's suggestions, also to make the story

more focused, we remove the SPSB3 E3 part and re-organize the title into “Proteostatic reactivation of the developmental transcription factor TBX3 drives BRAF/MAPK-mediated tumorigenesis”.

As I mentioned and documented in the initial version, the CRL5^{SPSB3}-mediated TBX3 degradation is surely an important and novel finding in the field. We have also gained more progresses regarding the biological significance of this mechanism as well as other involved modification events. We look forward to present the story in the near future. Hope you will understand the reorganization and provide more suggestions.

- Other comments:

- It is not always clear whether HEK293T (probably chosen because they are easier to transfect) or thyroid cancer cells (preferred models) were used in in vitro experiments (e.g., Fig 2). Please clarify, and if possible, prioritize observations that were identified in BRAFV600E mutant thyroid cancer cells, as their biology is disrupted by MAPK constitutive activation and different than other lineages.

Response: Thank the reviewer for the kind suggestion. Yeah, we conducted more *in vitro* function and mechanism studies in BRAFV600E-positive human PTC/ATC cell line K1, BCPAP, and 8505C, such as Co-IP experiments, co-stainings, Proximity Ligation Assays (PLA), as well as TBX3 half-life detections. Besides, we verified the functional correlation between Usp15 and Tbx3 through genetic studies in BRAFV600E-induced mPTC model, which reinforces the pathological significance of the current mechanisms. In the meanwhile, we chose HEK293T cell line as the tool cells to do general biochemical experiments according to most literatures.

- The choice of A375 melanoma cells for the experiments displayed in Figure 4 is highly questionable.

Response: Please let us explain the rationale. The reason we expand our biochemical and functional investigations into A375 cell is that we wonder whether the USP15-TBX3 regulations occur in different types of BRAFV600E-related tumors, even we take the thyroid physiological-pathological transformation as an example. Indeed, that is the case. The function of TBX3 downstream of BRAFV600E has also been addressed before^{3, 6}. During the revision, not only we find USP15 is subject to BRAF/MAPK/AP-1 cascade control, but also we observe the positively correlated protein abundances of USP15 and TBX3 in the large cohort of melanoma specimens (Fig 5h-5m). More impressively, once we sort out BRAFV600E mutated patients in the TCGA SKCM datasets, the positive correlation between USP15 with TBX3 or with BRAF is significantly increased (supplementary Fig 4h).

Therefore, the stability control of TBX3 by USP15 represents a critical proteostatic mechanism downstream of BRAF/MAPK-directed pathological transformation. Hope our findings will push forward adjuvant therapeutic strategy design against BRAF inhibitor resistant circumstances.

- Figure 7 is not very informative in its current form. I would remove the illustrations on embryogenesis, and possibly those on tumorigenesis, and focus instead on a

simplified version in which the opposing roles of CRL5 and USP15 on TBX3 regulation are displayed, based on the experimental data from this manuscript.

Response: As a group working on organ formation and tumor development, we always have a question whether embryonic programs could be reinitiated during tumorigenesis, especially when developmental factors, like TBX3, are critically required for tumorigenesis. Indeed, de-differentiation and acquirement of progenitor-like characteristics has been realized to be a common transformation during tumorigenesis. Regarding BRAF/MAPK activation caused tumors, BRAFV600E promotes de-differentiation of pigmented melanocytes and colon epithelial cells within genetic and organoid models^{7, 8}. Through thyroid carcinogenesis and in the mPTC model, BRAFV600E causes loss of lineage differentiation thus impaired iodine uptake, concurrently with excessive proliferation⁹. De-differentiation is also responsible for BRAF inhibitors therapy resistance, thus to understand the underlying mechanisms of de-differentiation is of great importance for targeting of BRAFV600E-related tumors.

Take thyroid as an example, a few lines of evidences point to the critical roles of UPS15-TBX3 axis in defining the differentiation states of epithelial cells during BRAF/MAPK activity-determined development and tumorigenesis. Firstly, p-Erks, Usp15, and Tbx3 are expressed parallelly, high in embryogenic progenitors and de-differentiated tumor cells, but low in differentiated cells. Secondly, knock-out of either Usp15 or Tbx3 leads to comparable over-differentiation in thyroid organogenesis. Thirdly, knock-out of either Usp15 or Tbx3 from BRAFV600E-induced thyroid carcinogenesis results in re-differentiation. Therefore, we conclude that high BRAF/MAPK/USP15 secures high abundance of TBX3 thus de-differentiated cell state at early embryonic and advanced oncogenic stages. While low BRAF/MAPK/USP15 determines low TBX3 thus differentiated cell state at postnatal tissue and initial transformation. These findings also support the rationale that BRAF/MAPK could initiate the tumorigenesis by re-activating conserved embryonic regulatory cascades, and provide new thoughts for BRAF/MAPK related therapy designs. Hope the reviewer finds this revision more focused and translational.

- The Introduction feels at times as systematic review instead of an introduction to the essential topics that are necessary to understand and justify the experimental approach adopted here.

Response: Sorry for not composing the introduction in a better flow, and we have revised it. We are very interested in whether de-differentiation process, critical for tumorigenesis, happens through recurrence of certain embryonic events and regulations. As a key and conserved transcriptional factor, TBX3 is involved in both embryonic development and tumorigenesis, especially BRAFV600E-associated tumorigenesis. Recently, BRAFV600E-induced transformation has been proved to involve lineage de-differentiation through multiple organ tumorigenesis^{7, 8}. We therefore wonder whether reactivation of main TBX3 functions and regulations accounts for BRAFV600E-induced de-differentiation. We have re-organized the introduction according to this flow, and hope it will help the readers to get into the

findings.

- There might be multiple roles of MAPK activation in development, but their main role in thyroid cancer pathogenesis is unequivocal: promoting cell proliferation. The authors touch on this topic here and there, but it would be helpful to include experiments that delineate the effects of MAPK which are dependent vs. independent of TBX3 biology.

Response: Regarding the relevance of our findings to BRAFV600E-driven thyroid cancer, we actually find TBX3 and USP15 are both indispensable during BRAFV600E-induced transformation in normal thyrocyte line Nthy-ori 3-1 as well as in early embryonic thyroid (E16.5) (Fig 7f and 7g), in addition to aforementioned molecular reprogramming and histological characterization of Tbx3 or Usp15 knockout mPTC (Fig 7). As expected, loss of Tbx3 significantly blocked BRAF/MAPK activity-caused proliferation increase (supplementary Fig 5e). Together with our findings from melanoma, we are more confident about the biological significance of USP15-TBX3 axis downstream of BRAF/MAPK activation.

- There are several grammatical errors throughout the manuscript, so I would recommend the authors to perform a thorough revision of the text.

Response: Thank you for the suggestion. We perform the revision thoroughly and get language support from some friends in the field.

Reviewer #3 (Remarks to the Author):

This paper claims to have identified a substrate of the deubiquitinase USP15 that may have relevance to cancer therapy. There is no conceptual advance, but the findings will be of interest to specialist readers and to the Pharma drug discovery world. It is an exhausting read with over 100 figure panels (I counted 66 in the main text). I think this is both unfair on the reviewer but also the prospective reader. It is incumbent on the authors to design well controlled and better quantitated experiments that distill down their essential message. I think they fail to do that here.

Response: Thank the reviewer for such helpful suggestions and we apologize for not presenting the findings in a more focused way. As the reviewer mentioned, we identify USP15 as the specific DUB for TBX3 stability control, which fills in the scientific void of TBX3 post-translational modification.

As a group working on organ formation and tumor development, we always have a question whether embryonic programs could be re-initiated during tumorigenesis, especially when developmental factors, like TBX3, are critically required for tumorigenesis. To understand the biological significance of USP15-TBX3 axis during BRAF/MAPK activity-determined development and tumorigenesis systematically, we compared the histological characteristics as well as molecular profilings between Tbx3 or Usp15 knockout developmental and tumorigenic models. We actually found that loss of either gene leads to comparable tumor redifferentiation, which parallels their overdifferentiation tendency during development. These findings support the rationale that BRAF/MAPK could initiate the tumorigenesis by re-activating conserved embryonic regulatory cascades, and provide new thoughts for BRAF/MAPK related therapy designs.

During the revision, we removed the SPSB3 part and focused on USP15-mediated TBX3 stabilization. We further confirm the interaction affinity and specificity between USP15 and TBX3 with GST-pull down, Proximity Ligation Assays (PLA)^{10, 11}, addition of USP11 control, and proteomic analysis of public proteome data, which all support the direct interaction and biological significance of USP15-TBX3 axis. Hope the reviewer find this revision more focused and convincing.

Major comments:

All experiments involving over-expression of USP15 are flawed in that they miss an essential control. To show specificity another USP family member DUB should be expressed at similar levels. Most obvious here would be the closest family members USP4 or USP11. Otherwise any effects of USP15 over-expression could be interpreted as simple mass action effects.

Response: Thank the reviewer for the suggestion. During the revision, we include the USP11 as the control. As shown in Supplementary Fig 1b, USP11 shows no detectable interaction with TBX3 even in the overexpression system, which contrasts

to the strong binding between USP15 and TBX3 (Fig 1b and 1c). Furthermore, USP11 could not remove the poly-ubiquitin chain from TBX3, as USP15 does (Fig 2a). These data support more about the interaction specificity between USP15 and TBX3.

Related revisions have been made on Figure 2 and Supplementary Figure 1, as well as on page 5 line 125 to line 127: The binding is highly specific, since USP11, the close family member of USP15, fails to bind with TBX3 (Supplementary Fig. 1b).

page 6 line 141 to line 144: We next verified whether USP15 targets TBX3 for deubiquitylation. As expected, ectopic expression of wild-type USP15 but not USP15^{C269A} or USP11 reduced the poly-ubiquitylation of TBX3, while USP15 removal augmented TBX3 poly-ubiquitylation (Fig. 2a, b).

The starting point is the co-IP of USP15 following over-expression of TBX3 in HEX293T cells. The over-expression must be enormous to visualise TBX3 and USP15 directly with what looks like a silver stained gel. In effect this experiment has been performed previously as part of a systematic analysis of protein-protein interactions, Bioplex, by the Harper/Gygi labs. In their case using either partner as bait- the other one was not identified. This has to make me somewhat skeptical of this foundational experiment.

Response: Yeah, the experiment at the starting point (Figure 1A) shows the result of Co-IP combined with Mass-spec. Basically, the Co-immunoprecipitates of Flag-tagged TBX3 was exchanged off the Flag-M2 beads and separated by SDS gel, silver staining. Then specific bands were cut out of the gel and subjected to Mass-spec analysis. Yeah, a large amount of TBX3 over-expressed lysates were used and enriched in the Co-IPs¹²⁻¹⁵, thus we were able to visualize the TBX3 and USP15 bands. This kind of protocol has been widely accepted for identifying interactome of different factors¹²⁻¹⁵.

The Bioplex platforms, especially Bioplex 3.0, built by the Harper/Gygi labs, have provided very powerful models of the human interactome to date^{16, 17}. Unfortunately, using either TBX3 or USP15 as bait, the other one was not identified through Bioplex system. For setting up the Bioplex system, the authors employed the HA system, while we used FLAG system in this study. We are not sure whether the efficacies of different systems could account part of the outcome distinction, since we noticed quite a few well-addressed interactions in different studies are not detected by the Bioplex system either, such like the interactions between USP15 and PARP1, USP9X and CEP131, USP22 and PPAR γ ^{10, 13, 18} (Figure 2).

To further validate the interaction between USP15 and TBX3, we conducted GST-pull down experiment using GST-USP15-U1 and purified Flag-TBX3, since we have narrowed down to U1 domain that mediates USP15 binding with TBX3. The result clearly confirms the interaction between USP15 and TBX3 (Fig 1h). Furthermore, we applied "Duolink[®] Proximity ligation assay (PLA)^{10, 11}" onto BRAFV600E mutated thyroid cancer cells, K1 cells.. Indeed, USP15 and TBX3 are colocalized within the cells, giving strong interaction signals (Fig 1c). Moreover, USP15-mediated stabilization of TBX3 is very important for BRAF/MAPK activity-induced tumorigenesis, thus USP15 and TBX3 are highly correlated with BRAFV600E signature and with poor

tumor prognosis within different types of BRAFV600E cancers, at both transcriptomic and proteomic levels (Fig 5b, d, l, m and supplementary Fig 4a, h). Taken together, we are more confident with the physical interaction and functional regulation between USP15 and TBX3 after these additional experiments as well as the phenotype analyses. Hope these revisions would help to resolve the reviewer's concerns.

[Redacted]

Figure 2. Interaction protein networks of USP15, USP9X and USP22 in Bioplex database.

Some of my other comments

Fig. 1d- does not show colocalisation- simply that both are in the nucleus.

Response: Thank the reviewer for pointing it out. We checked the direct interaction and colocalization between USP15 and TBX3 with the Duolink® Proximity Ligation Assays (PLA) ^{10, 11}. With the sensitive read out by Cy3 intensity, we detected

interaction mostly in the cytoplasm, also some in the nuclei. We also performed more IF co-stainings and provided more representative pictures in the revised version. Fig 1d has been replaced with the PLA detection result with statistical quantification. Hope the data are more informative.

Fig 1f- loading needs to be adjusted so that starting values, time 0, are more similar.

Response: The previous Fig 1f result is now presented as Figure 1i. To make the data more convincing, we have run the samples on the same gel and showed one exposure of the western blot, which further confirmed that USP15 knockdown shortens the half-life of TBX3.

Fig 1h -representative of several others. The MW of TBX3 is close to 95kD- yet the effect of USP15 overexpression is to reduce an entire ubiquitin smear, including the MW range below 95kD.

Response: We agree with the reviewer, the MW of TBX3 is below 95KD. Overexpression of USP15 leads to reduction of ubiquitin smear, mainly within the range above TBX3 size. When we zoom in, the smear around 55KD could be due to the IgG H chain left from the Co-IP experiments. These types of results could be shown as whole or partial membrane exposures. Herein, we re-organize the related results in Figure 2 to make the de-ubiquitylation results more appreciable.

Fig 6B quantitation does not seem to match the data presented. More sample need to loaded for -/- condition. How can you get an apparent increase in TBX3 after 2 hours cyclohex.

Response: We performed more rounds of MEFs culture and WB detections. Since the MEFs are isolated from Usp15-/- animals, the baseline expression of Tbx3 is pretty low and easy to be degraded.

Discussion of pre-existing USP15 literature is weak.

Response: As the reviewer suggested, we have replenished the discussion with more USP15 literatures. We discuss about the developmental defect of Usp15 ko mutant, as well as the potential involvement of Tbx3. And also, the function of USP15 across tumorigenesis and the significance of USP15-TBX3 regulation in TBX3 study was discussed. The translational potential of this finding is also explored. Hope the reviewer finds this revision more informative and convincing.

Related revisions have been made on discussion on page 16, line 421 to line 432: Multiple oncogenic pathways have been found to regulate TBX3 expression in occurrence of melanoma, lung, thyroid, breast and other types of cancers, here we define USP15 as a critical stabilizer for TBX3. Due to the conflicting functions of substrates, USP15 shows dual effects in tumorigenesis. For instance, USP15 copy number is increased in glioblastoma, breast cancer, and ovarian cancer, but decreased in pancreatic cancer. The current finding provides more evidence about the USP15 oncogenic function, especially during BRAF^{V600E} mutated cancers. Moreover, a few kinases participate in phosphorylation of TBX3, such as AKT3 at S720, cyclin A-CDK2 at either S190 or S354, p38 MAP kinase at S692, and AKT1 is also involved. Some of the

phosphorylation sites also have effect on the stability of TBX3, it will be interesting to check the functional correlation between different types of modifications.

References:

1. Mahon C, Krogan NJ, Craik CS, Pick E. Cullin E3 ligases and their rewiring by viral factors. *Biomolecules* **4**, 897-930 (2014).
2. Zhang P, *et al.* Targeting myeloid derived suppressor cells reverts immune suppression and sensitizes BRAF-mutant papillary thyroid cancer to MAPK inhibitors. *Nature Communications* **13**, (2022).
3. Boyd SC, *et al.* Oncogenic B-RAFV600E Signaling Induces the T-Box3 Transcriptional Repressor to Repress E-Cadherin and Enhance Melanoma Cell Invasion. *Journal of Investigative Dermatology* **133**, 1269-1277 (2013).
4. Li X, *et al.* TBX3 promotes proliferation of papillary thyroid carcinoma cells through facilitating PRC2-mediated p57KIP2 repression. *Oncogene* **37**, 2773-2792 (2018).
5. Corson LB, Yamanaka Y, Lai K-MV, Rossant J. Spatial and temporal patterns of ERK signaling during mouse embryogenesis. *Development* **130**, 4527-4537 (2003).
6. Peres J, Prince S. The T-box transcription factor, TBX3, is sufficient to promote melanoma formation and invasion. *Mol Cancer* **12**, 117 (2013).
7. Köhler C, *et al.* Mouse Cutaneous Melanoma Induced by Mutant BRaf Arises from Expansion and Dedifferentiation of Mature Pigmented Melanocytes. *Cell Stem Cell* **21**, 679-693.e676 (2017).
8. Reischmann N, Andrieux G, Griffin R, Reinheckel T, Boerries M, Brummer T. BRAFV600E drives dedifferentiation in small intestinal and colonic organoids and cooperates with mutant p53 and Apc loss in transformation. *Oncogene* **39**, 6053-6070 (2020).
9. Oh JM, Ahn B-C. Molecular mechanisms of radioactive iodine refractoriness in differentiated thyroid cancer: Impaired sodium iodide symporter (NIS) expression owing to altered signaling pathway activity and intracellular localization of NIS. *Theranostics* **11**, 6251-6277 (2021).
10. Sun X, *et al.* Loss of the receptors ER, PR and HER2 promotes USP15-dependent stabilization of PARP1 in triple-negative breast cancer. *Nat Cancer* **4**, 716-733 (2023).
11. Romani P, *et al.* Mitochondrial fission links ECM mechanotransduction to metabolic redox homeostasis and metastatic chemotherapy resistance. *Nature Cell Biology* **24**, 168-180 (2022).
12. Demirdizen E, *et al.* TRIM67 drives tumorigenesis in oligodendrogliomas through Rho GTPase-dependent membrane blebbing. *Neuro-Oncology* **25**, 1031-1043 (2023).
13. Li X, *et al.* USP9X regulates centrosome duplication and promotes breast carcinogenesis. *Nature Communications* **8**, (2017).
14. Ye B, *et al.* USP25 Ameliorates Pathological Cardiac Hypertrophy by Stabilizing SERCA2a in Cardiomyocytes. *Circulation Research* **132**, 465-480

- (2023).
15. Wei X, *et al.* Sigma - 1 receptor attenuates osteoclastogenesis by promoting ER - associated degradation of SERCA2. *EMBO Molecular Medicine* **14**, (2022).
 16. Schweppe DK, Huttlin EL, Harper JW, Gygi SP. BioPlex Display: An Interactive Suite for Large-Scale AP-MS Protein-Protein Interaction Data. *J Proteome Res* **17**, 722-726 (2018).
 17. Huttlin EL, *et al.* Dual proteome-scale networks reveal cell-specific remodeling of the human interactome. *Cell* **184**, (2021).
 18. Ning Z, *et al.* USP22 regulates lipidome accumulation by stabilizing PPAR γ in hepatocellular carcinoma. *Nature Communications* **13**, 2187 (2022).

REVIEWERS' COMMENTS

Reviewer #1 (Remarks to the Author):

The authors have addressed my comments satisfactorily.

Reviewer #4 (Replacing Reviewer #2, Remarks to the Author):

Overall, the findings and results are well demonstrated and should be published.

However, the hypothesis, rationale, and results are explained much better in the responses to reviewers than in the manuscript itself.

In the text, it is difficult to follow the story, what is the question, why the hypothesis, why the model or which one? Although the manuscript has improved with the additional paragraphs following the reviewer's comments. Very often, experiments are explained after the results are shown, or the models are explained long after their use, probably due to changes in the order of the results or the figures.

- There is no Fig 4i mentioned in the text, line 307.
- Why they used thyroid cancer as a model is not well explained.
- Cell lines.
 - They used cell lines not mentioned in the text or methods (for example in sup Fig 4b). Cal62 is a RAS-mutated thyroid cancer cell line and shows the same levels of USP15 and TBX3 protein levels as the other BRAF mutant thyroid cancer cell lines. This indicates that the modulation of those proteins may be due to activation of the MAPK pathway independently of the upstream mutation. This should be reflected in the text, or the cell line should be eliminated and must be discussed.
 - All the cell lines must show the driver mutation (BRAF, RAS, p53...) and tumor origin (thyroid, melanoma, lung...)
 - The K1 and 8505c thyroid cancer cell lines are explained in line 172 but they have been mentioned and used since the beginning of the results section, Line 111. Same with other cell lines. The authors need to go through the paper and explain the model the first time they use it or mention it in the text.
- MAPKi (BRAFi or MEKi). In the paragraph starting at line 213, they conclude without explaining the model or the inhibitors used, but they do in the second paragraph added. It does not flow. I miss a control in the western blots showing a correct MAPK pathway inhibition after treatment, such as pERK/ERK protein levels.
- The different patient cohorts are not well defined. Especially the one mentioned in line 247.
- In line 230, they show a western blot of the proteins of interest in embryonic thyroid, but they do not explain why. That is well explained in line 312.
- In line 349, they mentioned PAX8, which is not in the figure.
- In Figure 3f, band densitometry is not correct. IB: USP15 mPTC p30 and p50.
- Figures 3i and h are practically the same. I suggest consolidating the data in one graph.
- The authors should go throughout the paper, re-organize some parts, and explain the experimental models in each case

Reviewer #5 (Replacing Reviewer #3, Remarks to the Author):

In this manuscript entitled "Proteostatic reactivation of the developmental transcription factor TBX3 drives BRAF/MAPK-mediated tumorigenesis" the authors report a novel mechanism

by which the transcription factor TBX3 promotes tumorigenesis in BRAF/MAPK-driven cancers. They found that the protein stability of TBX3 depends on the USP15, which is upregulated by BRAF/MAPK and acts as a TBX3 deubiquitinase. The authors show that the BRAF/MAPK-USP15-TBX3 axis is critical for dedifferentiation during both embryonic development and tumorigenesis. The manuscript shows a wealth of biochemical, mouse in vivo data and correlative clinical data to support this proteostatic mechanism of TBX3 regulation and its role in tumorigenesis.

The manuscript reports a significant finding for the field. For the initially submitted manuscript the reviewers raised several technical and experimental concerns (e.g. the use of certain cell lines was questioned, the analysis of immunoblots and ICC and IHC images was not quantitative, the lack of a deubiquitinase control for USP15) and the lack of focus/accessibility of the initial manuscript. In the revised manuscript the authors have satisfactorily addressed all of these concerns.

20 February 2024

Re: Manuscript NCOMMS-23-08495A-Z

Thank you for your letter regarding the above submitted manuscript titled “Proteostatic reactivation of the developmental transcription factor TBX3 drives BRAF/MAPK-mediated tumorigenesis”, which was revised for publication on Nature Communications.

We are very grateful for the opportunity to have this manuscript published on your prestigious journal. We have undertaken revisions in response to the reviewers’ comments and would now like to submit a revised manuscript. The detailed responses and revisions are given below:

Reviewer #1 (Remarks to the Author):

The authors have addressed my comments satisfactorily.

Reviewer #4 (Replacing Reviewer #2, Remarks to the Author):

Overall, the findings and results are well demonstrated and should be published. However, the hypothesis, rationale, and results are explained much better in the responses to reviewers than in the manuscript itself.

In the text, it is difficult to follow the story, what is the question, why the hypothesis, why the model or which one? Although the manuscript has improved with the additional paragraphs following the reviewer's comments. Very often, experiments are explained after the results are shown, or the models are explained long after their use, probably due to changes in the order of the results or the figures.

Response: Thank the reviewer for the helpful suggestions and we apologize for having not organized the models in a better way. In the current version, we have made

necessary revisions and hope the manuscript get easier to follow.

- There is no Fig 4i mentioned in the text, line 307.

Response: Thank the reviewer for pointing this out. Yeah, there is no Fig 4i in the current version of manuscript, and we have deleted it from line 307.

- Why they used thyroid cancer as a model is not well explained.

Response: There are two reasons for choosing thyroid cancer as a model. First, de-differentiation has been observed in BRAF^{V600E}-induced transformation of different tissues, mainly thyroid cancer, melanoma, and colorectal cancer¹⁻⁵. Even BRAF^{V600E} is also the pivotal oncogenic driver within melanoma and colorectal cancer, the spontaneous thyroid cancer model we used here is the only mature transformation model accessible nowadays for studying tumorigenesis with this signature. The other reason is that our group has been working on development and tumorigenesis in the endocrine system for more than a decade and has set up a series of developmental and tumorigenic genetic models.

- Cell lines.

• They used cell lines not mentioned in the text or methods (for example in sup Fig 4b). Cal62 is a RAS-mutated thyroid cancer cell line and shows the same levels of USP15 and TBX3 protein levels as the other BRAF mutant thyroid cancer cell lines. This indicates that the modulation of those proteins may be due to activation of the MAPK pathway independently of the upstream mutation. This should be reflected in the text, or the cell line should be eliminated and must be discussed.

Response: We appreciate the constructive suggestion from the reviewer. Actually, expressions of USP15 and TBX3 were initially analyzed within the 152 patient tissue array, the majority of which are PTC and adjacent tissues, with a small number of other types of thyroid cancer tissues. When we checked the correlation between USP15 and TBX3, we observed positive correlation without distinguishing tumor molecular background. So we included thyroid cancer cell lines with different driver mutations to further check the correlation, such as TPC1 with RET rearrangement, Cal62 with RAS mutation, and we detected positive correlation between the two factors generally. When we expanded the detection also to other tissue derived cancers, we found the correlation is more reliable within BRAF/MAPK activation-induced or BRAF^{V600E} positive cancers. Together with our genetic study, we agree with the reviewer that the regulation is under MAPK activation background.

We supplemented related descriptions of included cell lines and addressed the point that “the modulation of those proteins could be due to MAPK pathway activation independently of the upstream mutation” in the results.

Related revisions have been made on results where cell lines were used for the first time on page 5, line 113 to line 118: Cell lines with different molecular backgrounds were included here, such as mouse embryonic fibroblasts (MEFs), BRAF^{V600E}-mutated papillary thyroid cancer (PTC) K1 cells and anaplastic thyroid cancer (ATC) 8505C cells, as well as an

estrogen receptor positive (ER+) breast cancer cell line MCF-7, and KRAS-mutated lung adenocarcinoma cell line A549, and showed different turnover rates.

As well as on page 11, line 271 to line 280: Indeed, the abundant expression of USP15 and TBX3 concurs apparently through BRAF^{V600E} positive cancer cells, such as melanoma cells A2058, OCM-1, A375, besides thyroid cancer cells K1, BCPAP, 8505C (Supplementary Fig. 4b). Additionally, USP15 and TBX3 also showed high expression within TPC-1 and Cal-62 cells with RET rearrangement or KRAS mutation respectively, indicating the modulation of these proteins may be due to BRAF/MAPK pathway activated with different upstream mutations. Take melanoma as another model where BRAF^{V600E}-induced BRAF/MAPK activation is also the most dramatic carcinogenic mutation, we found knock-down of USP15 resulted in significant TBX3 reduction (Supplementary Fig. 4c).

- All the cell lines must show the driver mutation (BRAF, RAS, p53...) and tumor origin (thyroid, melanoma, lung...)

Response: We supplemented related descriptions of included cell lines in the text, including the results, the legends, and methods.

- The K1 and 8505c thyroid cancer cell lines are explained in line 172 but they have been mentioned and used since the beginning of the results section, Line 111. Same with other cell lines. The authors need to go through the paper and explain the model the first time they use it or mention it in the text.

Response: We moved forward the description of the cell lines. We went through the paper and put explanation of the models forward or every first time we use them.

Related revisions have been made on results where cell lines were used for the first time on page 5, line 113 to line 118: Cell lines with different molecular backgrounds were included here, such as mouse embryonic fibroblasts (MEFs), BRAF^{V600E}-mutated papillary thyroid cancer (PTC) K1 cells and anaplastic thyroid cancer (ATC) 8505C cells, as well as an estrogen receptor positive (ER+) breast cancer cell line MCF-7, and KRAS-mutated lung adenocarcinoma cell line A549, and showed different turnover rates.

- MAPKi (BRAFi or MEKi). In the paragraph starting at line 213, they conclude without explaining the model or the inhibitors used, but they do in the second paragraph added. It does not flow. I miss a control in the western blots showing a correct MAPK pathway inhibition after treatment, such as pERK/ERK protein levels.

Response: We apologize for the inappropriate organization and description. We have moved forward the explanation of BRAFi or MEKi and added the p-ERK detection into Fig4a. As to the primary mPTC cell model, it's basically primary thyroid cancer cells FACSed by GFP from mPTC; Rosa26-mTmG line. The same model has been used in our previous TBX3 paper on *Nat Commun* as well⁶, so we corrected the description and added reference here.

Related revisions have been made on results on page 9, line 215 to line 222, as

well as on Figure 4a: Next, we wondered whether the continuously increased Usp15 through mPTC progressing induced by genetic BRAF^{V600E} incorporation was a tumor cell autonomous behavior. Primary mPTC cells were isolated from mPTC; Rosa26-mTmG reporter line, and subjected to BRAF inhibitor PLX4032, or ERK1/2 inhibitor SCH772984 treatment. Notably, Usp15 in the primary tumor cells is under BRAF/MAPK pathway control (Fig. 4a). Comparably, pharmacological inhibition of BRAF or ERK1/2 also significantly down-regulated USP15 expressions in BRAF^{V600E}-mutated human thyroid cancer cells (Fig. 4b and Supplementary Fig. 3a).

- The different patient cohorts are not well defined. Especially the one mentioned in line 247.

Response: Thank you for your careful reading and valuable suggestions. We have replenished the descriptions of different patient cohorts.

The patient cohort used on line 247 is a set of tissue array purchased from Shanghai Outdo Biotech company which includes 152 thyroid cancer and adjacent tissues, mostly PTC and with a small amount of other types of thyroid cancer.

To further check whether USP15-TBX3 modulation is molecular background-related, we received 93 pathological sections from Tianjin Medical University Cancer Institute & Hospital, independent of the 152 tissue array previously used. These 93 samples include 37 BRAF^{WT} and 56 BRAF^{V600E} mutated PTC tissues. We annotated related informations in the figure legends and supplementary materials.

Related revisions have been made on results on page 10, line 251 to line 254 as well as related figure legends, methods: Across the 152 tissue array samples composed mainly of PTC samples with adjacent tissues, as well as a small number of other types of thyroid cancer tissues, expression levels of USP15 and TBX3 are significantly increased and positively correlated with tumor progression⁶ (Supplementary Fig. 4a).

As well as on page 10-11, line 260 to line 262, and related figure legends, methods: We took advantage of another cohort of tissue samples consisted with PTC and the adjacent tissues, with accessible BRAF genetic information (n = 93), and analyzed via serial sectioning.

- In line 230, they show a western blot of the proteins of interest in embryonic thyroid, but they do not explain why. That is well explained in line 312.

Response: Thank you for pointing this out. We have highlighted the description of mPTC model in the revised manuscript. And “E16.5 embryonic thyroid from mPTC mice” here refer to embryonic thyroids isolated from mPTC mice normally.

Related description of the mouse model has been highlighted on results on page 8, line 190 to line 192: we first collected tumor tissues from mPTC model generated by crossing thyroid peroxidase *TPO-Cre* with *LSL-Braf*^{V600E} (Supplementary Fig. 2h).

- In line 349, they mentioned PAX8, which is not in the figure

Response: Thank you for your careful reading. Yeah, right, we didn't include PAX8

staining here and we corrected the description.

- In Figure 3f, band densitometry is not correct. IB: USP15 mPTC p30 and p50.

Response: Thank you for your pointing this out. We have corrected the band densitometry in figure 3f.

- Figures 3i and h are practically the same. I suggest consolidating the data in one graph.

Response: Thank you for the suggestion. The main information in figure 3h is to see whether knockout of Usp15 affects PTC development, so we compared the tumors from mPTC and mPTC/Usp15^{-/-}. For figure 3i, it mainly shows that removal of single allele of Usp15 (mPTC/Usp15^{+/-}) also represses PTC development. We keep the two panels separated since not every heterozygote in figure 3i has corresponding homozygote.

- The authors should go throughout the paper, re-organize some parts, and explain the experimental models in each case

Response: Thank the reviewer for the kind suggestion. We went through the paper and put explanation of the models forward or every first time we use them.

Reviewer #5 (Replacing Reviewer #3, Remarks to the Author):

In this manuscript entitled “Proteostatic reactivation of the developmental transcription factor TBX3 drives BRAF/MAPK-mediated tumorigenesis” the authors report a novel mechanism by which the transcription factor TBX3 promotes tumorigenesis in BRAF/MAPK-driven cancers. They found that the protein stability of TBX3 depends on the USP15, which is upregulated by BRAF/MAPK and acts as a TBX3 deubiquitinase. The authors show that the BRAF/MAPK-USP15-TBX3 axis is critical for dedifferentiation during both embryonic development and tumorigenesis. The manuscript shows a wealth of biochemical, mouse in vivo data and correlative clinical data to support this proteostatic mechanism of TBX3 regulation and its role in tumorigenesis.

The manuscript reports a significant finding for the field. For the initially submitted manuscript the reviewers raised several technical and experimental concerns (e.g. the use of certain cell lines was questioned, the analysis of immunoblots and ICC and IHC images was not quantitative, the lack of a deubiquitinase control for USP15) and the lack of focus/accessibility of the initial manuscript. In the revised manuscript the authors have satisfactorily addressed all of these concerns.

References:

1. Reischmann N, Andrieux G, Griffin R, Reinheckel T, Boerries M, Brummer T. BRAFV600E drives dedifferentiation in small intestinal and colonic organoids and cooperates with mutant p53 and Apc loss in transformation. *Oncogene* **39**, 6053-6070 (2020).
2. Romei C, *et al.* BRAFV600E mutation, but not RET/PTC rearrangements, is correlated with a lower expression of both thyroperoxidase and sodium iodide symporter genes in papillary thyroid cancer. *Endocrine Related Cancer* **15**, 511-520 (2008).
3. Knauf JA, *et al.* Targeted expression of BRAFV600E in thyroid cells of transgenic mice results in papillary thyroid cancers that undergo dedifferentiation. *Cancer Research* **65**, 4238-4245 (2005).
4. Feng L, Li J, Bu X, Zuo Y, Shen L, Qu X. BRAFV600E dictates cell survival via c-Myc-dependent induction of Skp2 in human melanoma. *Biochemical and Biophysical Research Communications* **524**, 28-35 (2020).
5. Kim M, *et al.* BRAFV600E Transduction of an SV40-Immortalized Normal Human Thyroid Cell Line Induces Dedifferentiated Thyroid Carcinogenesis in a Mouse Xenograft Model. *Thyroid* **30**, 487-500 (2020).
6. Zhang P, *et al.* Targeting myeloid derived suppressor cells reverts immune suppression and sensitizes BRAF-mutant papillary thyroid cancer to MAPK inhibitors. *Nature Communications* **13**, (2022).